# HIFα isoform specific activities drive cell-type specificity of *VHL*-associated oncogenesis

Joanna D.C.C. Lima [1,2], Madeleine Hooker[1,2], Ran Li [2], Ayslan B. Barros[1,2], Norma Masson[1,2], Christopher W. Pugh[2], David R. Mole [2], Julie Adam [1,2], Peter J. Ratcliffe [1,2,3,4] ✉ & Samvid Kurlekar [1,2,4] ✉

Cancers arising from dysregulation of generally operative signaling pathways are often tissue specific, but the mechanisms underlying this paradox are poorly understood. Based on striking cell-type specificity, we postulated that these mechanisms must operate early in cancer development and set out to study them in a model of von Hippel Lindau (VHL) disease. Biallelic mutation of the VHL ubiquitin ligase leads to constitutive activation of hypoxia inducible factors HIF1A and HIF2A and is generally a truncal event in clear cell renal carcinoma. We used an oncogenic tagging strategy in which *VHL*-mutant cells are marked by tdTomato, enabling their observation, retrieval, and analysis early after *VHL*-inactivation. Here, we reveal markedly different consequences of HIF1A and HIF2A activation, but that both contribute to renal cell-type specific consequences of *VHL*-inactivation in the kidney. Early involvement of HIF2A in promoting proliferation within the proximal tubular epithelium supports therapeutic targeting of HIF2A early in VHL disease.

The von Hippel Lindau (*VHL*) tumor suppressor gene (TSG) is a classical TSG which is mutated in a highly tissue-restricted range of tumors[1,2]. Its biallelic inactivation is almost universally the truncal driver for both syndromic and sporadic clear cell renal cell carcinoma (ccRCC)[3], the most common malignancy observed in the kidney[4]. ccRCC is generally thought to arise from the proximal tubule (PT) of the renal tubular epithelium (RTE) suggesting that *VHL* inactivation has a unique oncogenic potential in this cell lineage[5,6]. Despite this cell-specificity, *VHL* is expressed ubiquitously and encodes the recognition subunit of an E3 ubiquitin ligase complex that is best characterized as targeting the α-subunits of hypoxia inducible transcription factors (HIFα) for oxygen-dependent degradation[7]. Following *VHL* inactivation in PT cells, HIFα is constitutively stabilized leading to upregulation of its transcriptional targets irrespective of oxygen levels[8].

The tissue-specificity of *VHL*-associated cancer suggests that the oncogenic potential of HIFα activation in this setting is itself tissue-specific, leading to a major focus on defining the underlying mechanisms. HIFα has two principal isoforms, HIF1A and HIF2A which have partially overlapping sets of transcriptional targets[9–12]. While HIF1A is expressed widely, HIF2A is expressed at variable levels in a more limited number of tissues[13]. Although many studies support the existence of contrasting roles for HIF1A and HIF2A in the development of ccRCC, most have focused on systems representing mature or even late-stage cancer. For instance, multiple studies have identified HIF1A as unfavorable and HIF2A as favorable to the growth of ccRCC cells as xenografts[9,14–16]. The loss of chromosome 14q, which harbors the *HIF1A* locus, associates with worse prognosis in ccRCC, potentially supporting an anti-oncogenic role for HIF1A in this setting[17,18]. In contrast, and consistent with a pro-oncogenic role for HIF2A, HIF2A specific antagonists retard the growth of ccRCC tumors in patients and ccRCC-derived xenografts[19–21]. An explanation for these effects has been suggested by the observation that HIF2A co-operates with the RTE-specific transcription factor PAX8 to upregulate cyclin D1 in ccRCC cells[22].

[1]Ludwig Institute for Cancer Research, Nuffield Department of Medicine, University of Oxford, Oxford, UK. [2]Nuffield Department of Medicine, University of Oxford, Oxford, UK. [3]The Francis Crick Institute, 1 Midland Road, London, UK. [4]These authors jointly supervised this work: Peter J. Ratcliffe and Samvid Kurlekar. ✉e-mail: peter.ratcliffe@ndm.ox.ac.uk; samvid.kurlekar@ndm.ox.ac.uk

Crucially, these insights into the roles of HIFα isoforms in advanced disease do not necessarily inform on the earliest stages of ccRCC development. Biological understanding of such events is important, particularly in view of the potentially decades-long latency before *VHL*-null cancer cells develop into clinically detectable tumors[3,23]. This is also of potential therapeutic importance following the development of isoform-specific inhibitors of HIFα, such as the HIF2A-specific antagonist Belzutifan[24,25]. One means of interrogating the early effects of HIFα activation is by co-inactivation of each of the genes encoding HIFα isoforms with *Vhl*. Interestingly, when this has been done, the effects of HIFα isoform activation appear more complex[11,12,26].

In a mouse model deploying concomitant *Vhl* and *Trp53* and/or *Rb1* inactivation, tumor incidence was delayed or completely prevented by co-deletion of either *Hif1a* (encoding HIF1A) or *Epas1* (encoding HIF2A), suggesting that both HIFα isoforms are necessary for oncogenesis in this setting[11,12]. However, this work involved the use of oncogenic mutations that are rarely seen in ccRCC[27] and does not resolve actions of HIFα isoforms at the cellular level. Studying the earlier effects of *Vhl* inactivation has also been limited by a technological inability to identify and assay *VHL*-null cells in tissue precisely and in advance of morphological abnormality. To overcome this limitation, we have developed an 'oncogenic cell tagging' mouse model in which *Vhl* recombination is linked structurally to tdTomato expression[28]. This model allows for the accurate tagging and retrieval of cells that have undergone *Vhl* inactivation and subsequent analysis by immunohistochemistry and single-cell RNA sequencing (scRNA-seq)[28].

Earlier analyses with this model using *Pax8-CreERT2*[29] to drive *Vhl* inactivation in the RTE from the PT of the cortex to the collecting ducts of the papilla revealed cell-type specific patterns of proliferation and elimination which suggested that the tissue-specificity of *Vhl*-associated oncogenesis might be coded by cellular tolerance and adaptation to *Vhl* inactivation that is manifest very early during ccRCC evolution[28]. However, the mechanisms underlying these processes have not yet been studied.

Here, we have examined the HIFα-dependence of cellular responses and gene expression after *Vhl* inactivation in the RTE in vivo, by coupling the oncogenic cell tagging model of *Vhl* inactivation with conditionally inactivated alleles for *Hif1a* and/or *Epas1* (which encodes HIF2A). Our findings resolve the roles of different HIFα isoforms in this setting and reveal marked cell-type specific effects along the RTE.

## Results

### Combined inactivation of *Hif1a* and *Epas1* in tdTomato-tagged *Vhl*-null cells

To define tissue-specific actions of biallelic *Vhl* inactivation, an oncogenic cell tagging *Vhl* allele[28] (*Vhl^{pjr.fl}*; Supplementary Fig. 1a) was combined with either a constitutively inactivated (*Vhl^{jae.KO}*) or wild-type (*Vhl^{wt}*) second allele of *Vhl*. Biallelic *Vhl* inactivation was induced specifically in the RTE by activation of an RTE-restricted Cre recombinase[29] in *Vhl^{jae.KO/pjr.fl}*; *Pax8-CreERT2* ('VKO') mice, following tamoxifen administration (Supplementary Fig. 1b). Phenotypes in these mice were compared to those in *Vhl^{wt/pjr.fl}*; *Pax8-CreERT2* ("ConKO") mice (Supplementary Fig. 1b), in which Cre recombinase induced monoallelic *Vhl* inactivation. This comparison mimics events in the clinical syndrome of VHL disease where one defective allele is inherited, and the other is subsequently inactivated somatically[30].

Mice of each genotype were then bred with those carrying conditional 'floxed' alleles for *Hif1a* and/or *Epas1* to generate mice of the following genotypes: *Vhl^{jae.KO/pjr.fl}*; *Hif1a^{fl/fl}*; *Pax8-CreERT2* ("VHKO"), *Vhl^{jae.KO/pjr.fl}*; *Epas1^{fl/fl}*; *Pax8-CreERT2* ("VEKO"), *Vhl^{jae.KO/pjr.fl}*; *Hif1a^{fl/fl}*; *Epas1^{fl/fl}*; *Pax8-CreERT2* ("VHEKO"), *Vhl^{wt/pjr.fl}*; *Hif1a^{fl/fl}*; *Pax8-CreERT2* ("ConHKO"), *Vhl^{wt/pjr.fl}*; *Epas1^{fl/fl}*; *Pax8-CreERT2* ("ConEKO"), and *Vhl^{wt/pjr.fl}*; *Hif1a^{fl/fl}*; *Epas1^{fl/fl}*; *Pax8-CreERT2* ("ConHEKO").

The cell tagging system was linked only to the *Vhl^{pjr.fl}* allele, so first we tested, using PCR analyses of genomic DNA, if tdTomato-positivity

(reflecting *Vhl^{pjr.fl}* recombination) also marked concomitant *Hif1a* and/or *Epas1* deletion in the relevant genotypes. tdTomato-positive cells from VHKO and VHEKO, but not VEKO kidneys, carried the recombined form (*Hif1a^{KO}*) of the *Hif1a* allele (Supplementary Fig. 1c). Similarly. tdTomato-positive cells from VEKO and VHEKO but not VHKO kidneys carried the recombined form (*Epas1^{KO}*) of the *Epas1* allele (Supplementary Fig. 1c). Although unrecombined forms of the *Hif1a* and *Epas1* alleles (*Hif1a^{fl}* and *Epas1^{fl}*) were also detected, signals were very weak (Supplementary Fig. 1c). Taken together, these results indicate that the large majority of tdTomato-positive cells in VHKO, VEKO, and VHEKO mice are also *Hif1a*-null, *Epas1*-null, and *Hif1a/Epas1*-null, respectively.

### Contrasting effects of HIFα isoforms on cell-type specific responses to *Vhl* inactivation in the kidney

Our previous work on the effects of *Vhl* inactivation using this model demonstrated a substantial time-dependent loss of tdTomato-positive cells in the renal papilla following biallelic inactivation of *Vhl*[28]. To assess the actions of HIFα on this phenotype, we induced combined recombination of *Vhl* and genes encoding either or both HIFα isoforms *Hif1a* and *Epas1* and compared the number of tdTomato-positive cells in the papilla across all mouse genotypes (VKO, VHKO, VEKO, VHEKO) using identical methodology to that used previously[28]. In VKO mice, a striking loss ($p = 0.020$) of *Vhl*-null tdTomato-positive cells was observed in the papillae of kidneys examined at 4–12 months *versus* 1–3 weeks after *Vhl* inactivation (Fig. 1a, b). This reduction was abrogated in VHEKO ($p > 0.999$) or VHKO mice ($p = 0.311$), in which tdTomato-positive cells were *Hif1a*-null. In contrast, the reduction in papillary cells was maintained in VEKO mice ($p = 0.006$), in which tdTomato-positive cells were *Epas1*-null (Fig. 1a, b). No statistical difference was observed in any of the ConKO, ConHKO, ConEKO, and ConHEKO mice (Supplementary Fig. 2a, b). This data demonstrates that the loss of *Vhl*-null RTE cells in the renal papilla requires HIF1A, but not HIF2A.

In contrast to the survival disadvantage experienced by *Vhl*-null cells in the renal papilla, we have shown that *Vhl*-null PT cells in the renal cortex and outer medulla survive for at least 12 months post recombination and increase in number owing to transient proliferation[28]. Therefore, we sought to assess the HIFα-dependence of these phenotypes. No loss of tdTomato-positive cells was observed over time in the renal cortex and outer medulla in VHKO, VEKO, and VHEKO mice, indicating that neither HIFα isoform is necessary for the survival of *Vhl*-null cells. However, the gain in number of tdTomato-positive cells that was observed in VKO mice ($p = 0.016$) was absent in VHEKO and VEKO mice ($p > 0.999$) and did not attain statistical significance ($p = 0.335$) in VHKO mice (Fig. 1c, d). As with the renal papilla, none of the *Vhl*-competent genotypes (ConKO, ConHKO, ConEKO, and ConHEKO) exhibited significant changes in tdTomato-positive cell number over time (Supplementary Fig. 2c, d). This indicates that the full extent of *Vhl*-dependent proliferation depends on the integrity of both HIFα isoforms.

To pursue this, we performed further analyses of the tdTomato-tagged cells. A tdTomato-positive cell will create two neighboring tdTomato-positive cells upon cell division. We therefore hypothesized that division and survival of tdTomato-positive cells should be reflected in a greater number of tdTomato-positive immediate neighbors at 4–12 months when compared to 1–3 weeks after *Vhl* inactivation. Over 98% of tdTomato-positive cortical and outer medullary cells had at least one other cell, tagged or not tagged, within an internuclear distance of 16 μm (Supplementary Fig. 3a). This distance was thus chosen to denote the 'immediate neighborhood' for tdTomato-positive cells. The frequency distributions of, and the mean number of other tdTomato-positive cells within the neighborhood of each tdTomato-positive cell were then compared over time across all genotypes to analyze proliferation of cells in a manner not dependent on levels of initial recombination induced in our model. Early (1–3 weeks) after *Vhl* inactivation, tdTomato-positive cortical and outer medullary cells in

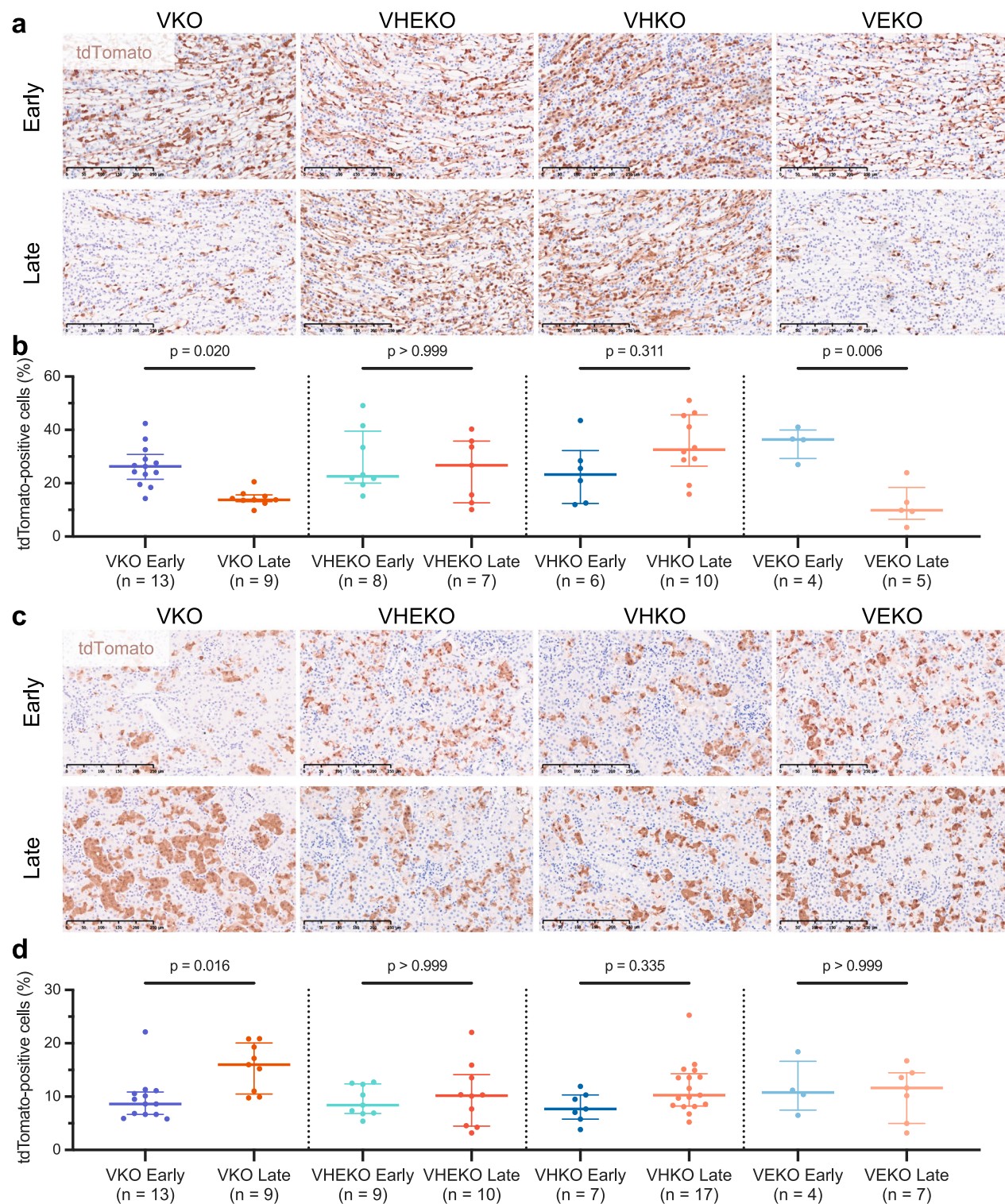

**Fig. 1 | Contrasting response to HIFα activation in different renal regions.**
Representative tdTomato IHC counterstained with hematoxylin in the renal papilla (**a**) or renal cortex and outer medulla (**c**) of VKO, VHKO, VEKO, and VHEKO mice given 5 × 2 mg tamoxifen and harvested early (1–3 weeks) or late (4–12 months) following recombination. Scale bar denotes 250 μm; ×20 magnification. Automated quantification (see "Methods") of the proportion of cells that are tdTomato-positive in the renal papilla (**b**) or renal cortex and outer medulla (**d**). Pairwise comparisons by Kruskal-Wallis test with Dunn's correction. Data are presented as median values, with the inter-quartile range indicated by error bars. The number of biological replicates for each condition is indicated.

VKO, VHKO, VEKO, and VHEKO mice exhibited a similar spatial distribution, with >75% having one or no tdTomato-positive neighbors within their neighborhood (Supplementary Fig. 3b, c). An increase in the number of tdTomato-positive neighbors was observed over time in VKO cells ($p = 0.007$ and $p = 0.003$ for frequency distributions and medians, respectively) that was absent in VHEKO ($p > 0.999$ in both analyses) and VEKO ($p > 0.999$ and $p = 0.719$) cells (Fig. 2a, b). The number of tdTomato-positive neighbors increased over time in VHKO cells ($p = 0.034$ and $p < 0.001$), but to a lower extent than in VKO cells (Fig. 2a, b). In contrast, none of the *Vhl*-competent genotypes (ConKO,

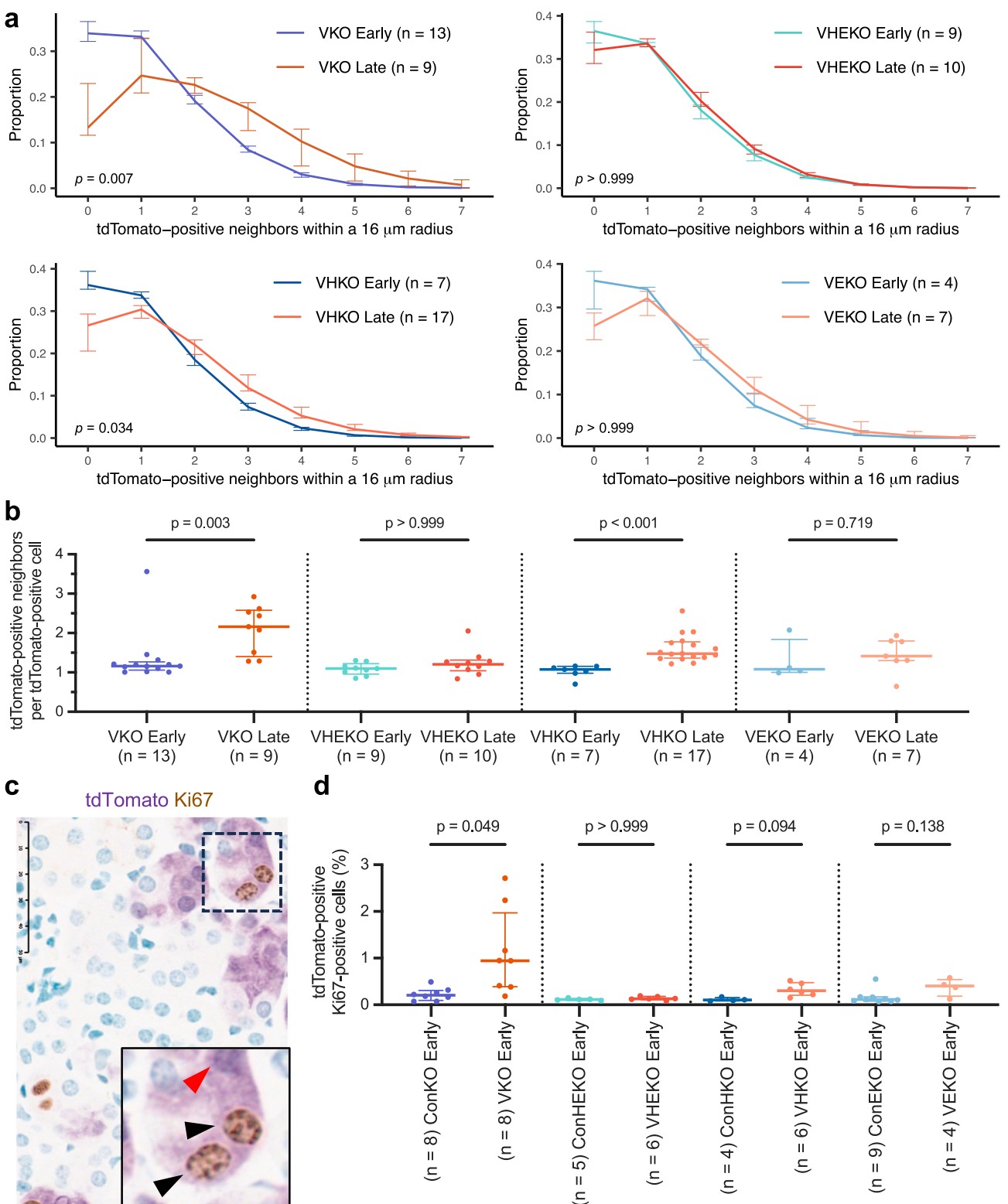

**Fig. 2 | *Vhl*-null cortical and outer medullary cells expand and proliferate in a HIFα-dependent manner.** Frequency distribution (**a**), and mean number (**b**) of tdTomato-positive neighbors of tdTomato-positive cells within a 16 μm radius in the cortex and outer medulla of VKO, VHEKO, VHKO, and VEKO mice harvested early or late after *Vhl* inactivation. **c** Representative dual IHC for Ki67 (brown) and tdTomato (purple) counterstained with methyl green. Scale bar denotes 50 μm; ×40 magnification. Area shown in the inset (black rectangle) illustrates tdTomato-positive Ki67-positive (black arrows) and tdTomato-positive Ki67-negative (red arrow) cells. **d** Automated quantification (see "Methods") of the proportion of tdTomato-positive cells that are also Ki67-positive in cortex and outer medulla of ConKO, VKO, ConHKO, VHKO, ConEKO, VEKO, ConHEKO, and VHEKO mice harvested early after recombination. **a, b, d** Data are presented as median values, with the inter-quartile range indicated by error bars. The number of biological replicates for each condition indicated. **a** Frequency distributions compared using Wilk's lambda statistic after testing by multivariate analysis of variance (MANOVA). **b, d** Pairwise comparisons by Kruskal-Wallis test with Dunn's correction.

ConHKO, ConEKO, and ConHEKO) exhibited significant changes in tdTomato-positive cell number over time (Supplementary Fig. 3d).

We then tested whether the HIFα-dependent differences in accumulation of *Vhl*-null cells in the cortical and outer medullary renal tubules at 4–12 months were associated with HIFα-dependent differences in proliferation early (1–3 weeks) after *Vhl* inactivation. We performed dual immunohistochemistry for tdTomato and the cell cycle marker Ki67 and quantified the proportion of tdTomato-positive cells that stained positive for Ki67 in ConKO, VKO, ConHKO, VHKO, ConEKO, VEKO, ConHEKO, and VHEKO mice early (1–3 weeks) after *Vhl* inactivation (Fig. 2c). The proportion of tdTomato-positive cells that were also Ki67-positive was increased in VKO *versus* ConKO cells ($p = 0.049$). This difference was absent ($p > 0.999$) when comparing VHEKO to ConHEKO cells (Fig. 2d). This increase was smaller and did not attain statistical significance in comparisons of VHKO *versus* ConHKO ($p = 0.094$) and VEKO *versus* ConEKO cells ($p = 0.138$) (Fig. 2d).

Taken together, these results indicate that in contrast with cells in the renal papilla, *Vhl*-null RTE cells in the cortex and outer medulla are tolerant of HIFα-activation and that they enter the cell cycle early after *Vhl* inactivation and multiply, in a manner dependent on the integrity of both HIFα isoforms.

## Isoform-specific HIFα-dependent gene expression in *Vhl*-null PT cells

To better understand the contributions of different HIFα isoforms to this behavior, we performed scRNA-seq on FAC-sorted tdTomato-positive cells isolated from VHKO, VEKO, and VHEKO mice 4–12 months after *Vhl* inactivation. Data was obtained from 147,045 high-quality cells from 7 male and 5 female mice (4 mice per genotype), with an average of $12,254 \pm 6438$ (SD) cells per mouse, $4933 \pm 2037$ (SD) mapped reads per cell, and $1594 \pm 408$ (SD) genes detected per cell. To facilitate comparison of this scRNA-seq data amongst the genotypes we first classified cells by nephron cell type using specific marker genes obtained from published scRNA-seq and anatomical studies in the kidney[31–35]. 95.7% of cells were assigned as PT S1, S2, or S3 proximal tubular cell types, consistent with the PT-restriction of *Pax8-CreERT2* under these experimental conditions[28].

After performing dimension reduction, we visualized the data by uniform manifold approximation (UMAP). Cells from different mice of the same genotype (VHKO, VEKO or VHEKO) occupied overlapping positions on the UMAP plot, indicating reproducible patterns of gene expression within genotypes (Supplementary Fig. 4a). Cells of the same nephron cell types were also positioned together in UMAP space, supporting the accuracy of cell type assignment (Supplementary Fig. 4b). Interestingly, the UMAP analyses also revealed a previously observed dichotomy in gene expression in PT cells that was independent of cell type (PT S1, S2, S3)[28]. This dichotomy in PT gene expression, which we have termed 'PT Class A' and 'PT Class B', was driven by the same genes (provided in Supplementary Data 1) as was observed previously and it was present in all HIFα-inactivated genotypes (VHKO, VEKO, and VHEKO), indicating that it is independent of the status of HIFα-encoding genes (Supplementary Fig. 4c, d). The existence of this dichotomy in vivo was confirmed by visualization of the cell-specific expression of genes characterizing PT Class A and Class B cells by in situ RNA hybridization in all HIFα-inactivated genotypes (Supplementary Fig. 4e). Taken together, cell type and class provided six 'PT identities' within each of which gene expression was compared. To enable these genotypic differences to be compared to those in cells bearing inactivation of *Vhl* alone (VKO) and to a *Vhl*-competent 'control' (ConKO) cells, we amalgamated data, obtained in an identical manner but reported previously, for ConKO and VKO cells[28]; these two datasets manifest a similar composition of the six PT identities (Supplementary Fig. 4f), facilitating amalgamation and comparison.

*Vhl* inactivation has heterogenous and cell-specific transcriptomic effects in the RTE[28] (Supplementary Data 2). To decipher the

contributions of HIFα isoforms to these effects precisely, we analyzed gene expression changes across ConKO, VKO, VHKO, VEKO, and VHEKO genotypes separately in cells of each PT identity. We first used Louvain clustering on this amalgamated data to compare the magnitude of differences. VHKO cells, bearing *Hif1a* deletion, were always present in the same clusters as VKO cells (Fig. 3a). In contrast, VEKO cells, bearing *Epas1* deletion, did not occupy the same clusters as VKO cells (Fig. 3a). This pattern was seen in all six PT identities. It was also evident in the UMAP plots of each PT identity; while VHKO cells (brown) overlapped with VKO cells (orange), VEKO cells (green) did not (Fig. 3a). These observations indicate that *Epas1* inactivation had a greater impact than *Hif1a* inactivation on the transcriptional program over 4–12 months after *Vhl* inactivation in all PT identities. VHEKO cells (in which both HIFα isoforms were inactivated) did not cluster together with VKO cells, but somewhat surprisingly neither did they occupy the same clusters as ConKO cells (Fig. 3a), suggesting that HIFα isoform gene co-inactivation did not return *Vhl*-null PT cells to *Vhl*-competent gene expression patterns completely. However, we are unable to determine the extent to which this represents truly HIFα-independent effects of *Vhl* inactivation, as opposed to basal activity of HIFα in ConKO cells or low levels of active HIFα isoforms retained in VHEKO cells.

To define the specific genes underlying these HIFα-dependent transcriptomic changes, we integrated single-cell data by 'pseudo-bulking' (see "Methods") and performed differential expression analysis across the different genotypes in each PT identity separately. We first sought to define the overall dependence on HIFα of changes induced by *Vhl* inactivation. We defined HIFα-dependent genes as those whose regulation following *Vhl* inactivation was reversed significantly ($p < 0.05$) and by at least 50% of the $\log_2$-fold change by *Hif1a* and/or *Epas1* inactivation. We then visualized the degree of HIFα-dependence for genes regulated by *Vhl* inactivation by comparing differential gene expression in VHEKO *versus* VKO cells (effect of HIFα codeletion in *Vhl*-null cells) against that in VKO *versus* ConKO cells (effect of *Vhl* inactivation) in cells of each PT identity (Supplementary Fig. 5a). Genes that were classified as being HIFα-dependent exhibited almost total anti-correlation (Spearman's $\rho < -0.8$) in this analysis. Notably, genes that failed to meet our threshold for HIFα-dependence also displayed an anti-correlation (Spearman's $\rho < -0.5$), suggesting some effect of HIFα co-deletion (Supplementary Fig. 5a). Furthermore, we did not identify any distinct set of genes that were completely unaffected by HIFα co-deletion. Coupled with the possibility that incomplete recombination of *Hif1a* and *Epas1* alleles (see Supplementary Fig. 1c) underestimates the magnitude of differential expression in VHEKO *versus* VKO cells in our analysis, these observations indicate that the *Vhl*-regulated transcriptome in cells of each PT identity is very largely HIFα-dependent.

To define the isoform-specificity of HIFα-dependent gene regulation, we classified genes based on whether their *Vhl*-dependent regulation was significantly reduced—by at least 50% of the $\log_2$-fold change—upon inactivation of individual HIFα isoforms. Genes were classified as 'HIF1A alone' if this reduction occurred with *Hif1a* but not *Epas1* inactivation, 'HIF2A alone' if the reduction occurred with *Epas1* but not *Hif1a* inactivation, 'HIF1A or HIF2A' if regulation was reduced by inactivation of both *Hif1a* and *Epas1* individually, or 'HIF1A and HIF2A combined' if regulation was only reduced when both *Hif1a* and *Epas1* were inactivated concomitantly but not individually (Supplementary Fig. 5b). The isoform-specific regulation of some *Vhl*-dependent genes was also validated by in situ RNA hybridization (Supplementary Fig. 6). Consistent with the clustering analysis above, this showed that more genes were regulated after *Vhl* inactivation in a HIF2A-specific than in a HIF1A-specifc manner (Fig. 3b). Strikingly, this pattern was seen in all PT identities (Fig. 3b), even though the complement of *Vhl*-dependent and HIFα-dependent genes was different in each PT identity (data provided as a resource in Supplementary Data 2).

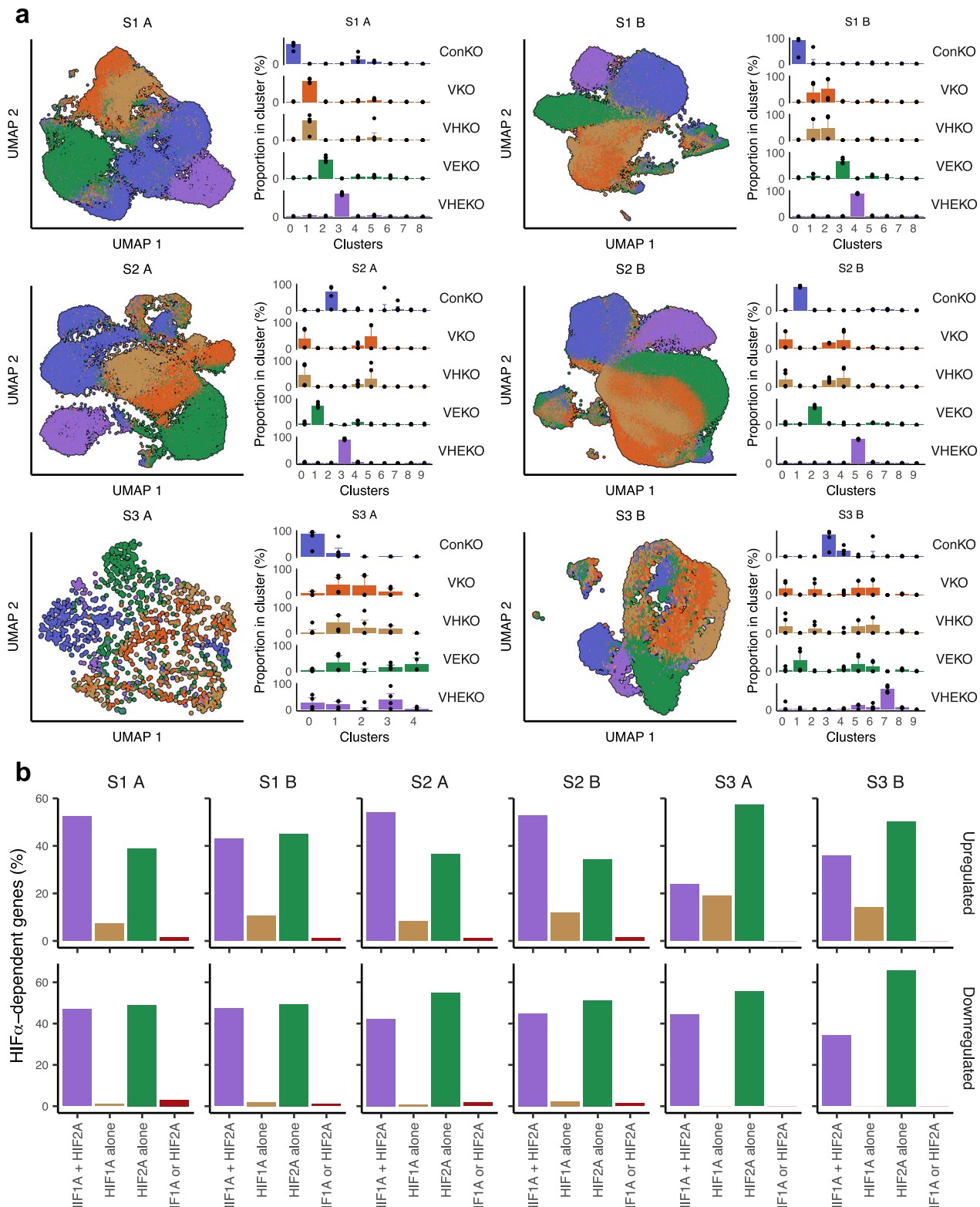

**Fig. 3 | HIFα-dependent gene expression in *Vhl*-null cells. a** UMAP plots depicting cells of the six PT identities from ConKO (blue), VKO (orange), VHKO (brown), VEKO (green), and VHEKO (purple) mice. Distribution of cells from each genotype across Louvain clusters are shown alongside each UMAP plot. Data are presented as median values, with the inter-quartile range indicated by error bars. *n* = 4 mice per genotype. **b** Isoform specificity of HIFα-dependent upregulation (top) or downregulation (bottom) of genes following *Vhl* inactivation in each PT identity. Bar charts depict the proportion of HIFα-dependent genes whose regulation was reversed by individual deletion of either *Hif1a* or *Epas1* ('HIF1A or HIF2A'; maroon), *Hif1a* but not *Epas1* ('HIF1A alone'; brown), *Epas1* but not *Hif1a* ('HIF2A alone'; green), or only by combined *Hif1a* and *Epas1* deletion ('HIF1A + HIF2A'; purple).

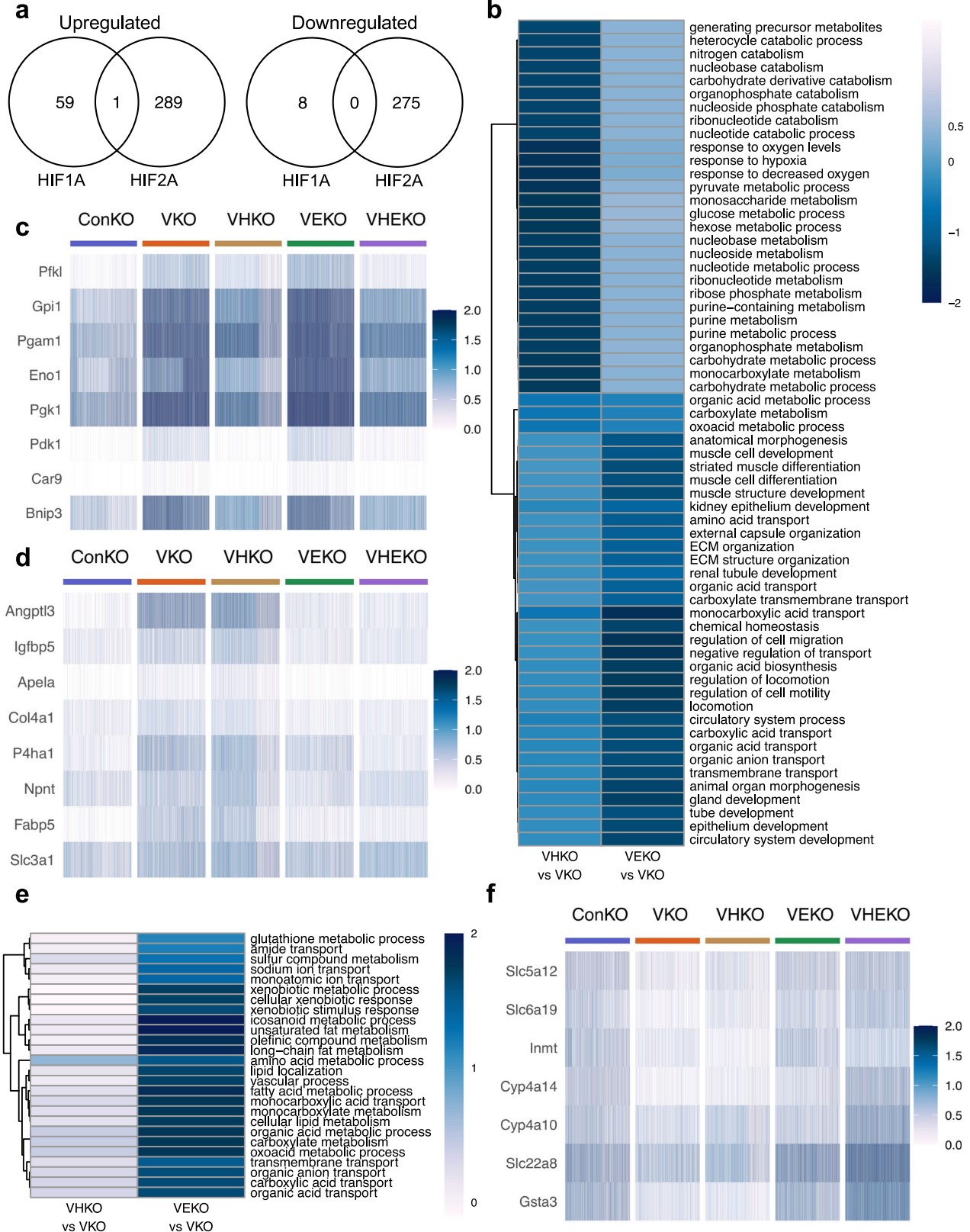

## Contrasting biological actions of HIF1A- and HIF2A-specific gene regulation

Though the genes regulated by HIFα varied across PT identities (Supplementary Data 2), the HIFα isoform-specificity of the regulation of each gene was retained whenever regulation by HIFα was manifest (Fig. 4a and Supplementary Fig. 5b). Thus, if a gene was regulated specifically by HIF1A in one PT identity, it was almost never regulated specifically by HIF2A in any other PT identity, and vice versa. This enabled HIFα isoform-specific genes across PT identities to be analyzed collectively and to be functionally characterized by reference to Gene Ontology (GO). Strikingly, no overlap was observed between GO terms over-represented significantly ($p < 0.01$) within HIF1A- and

**Fig. 4 | HIF1A and HIF2A regulate distinct biological processes in *Vhl*-null cells.** **a** Venn diagrams depicting intersections between genes whose *Vhl*-dependent upregulation (left) or downregulation (right) in any PT identity is reversed specifically by *Hif1a* deletion ('HIF1A') or specifically by *Epas1* deletion ('HIF2A'). 'HIF1A' and 'HIF2A' gene sets have been pooled across all PT identities. Only one or none of the genes are up- or downregulated specifically by different HIFα isoforms in different PT identities. **b** Gene Ontology terms for biological processes that are significantly over-represented ($p < 0.01$; one-sided Fisher's exact test with multiple testing correction by false discovery rate) within groups of genes whose *Vhl*-dependent upregulation is reversed specifically by *Hif1a* or *Epas1* deletion. Terms are ordered and tiles are colored by the average $\log_2$-fold changes for HIFIA- or

HIF2A-dependent genes that are members of each GO term in VHKO *vs* VKO or VEKO *vs* VKO mice. Single-cell heatmaps depicting scaled expression of selected genes whose *Vhl*-dependent upregulation is specifically reversed by *Hif1a* deletion (**c**), or *Epas1* deletion (**d**). **e** Gene Ontology terms for biological processes that are significantly over-represented ($p < 0.01$; one-sided Fisher's exact test with multiple testing correction by false discovery rate) within genes whose *Vhl*-dependent downregulation is reversed specifically by *Epas1* deletion ('HIF2A'). Terms are ordered and tiles are colored by the average $\log_2$-fold changes for HIFIA- or HIF2A-dependent genes that are members of each GO term in VHKO *vs* VKO or VEKO *vs* VKO mice. **f** Single-cell heatmaps depicting scaled expression of selected genes whose *Vhl*-dependent downregulation is specifically reversed by *Epas1* deletion.

HIF2A-specific upregulated genes (Fig. 4b), indicating that the two HIFα isoforms regulated distinct biological processes. In an effort to define common functions encompassed by the over-represented GO terms, some with redundant annotation, we clustered the GO terms based on similarities in the genes that drove their over-representation (Supplementary Fig. 7–9). This process defined glycolysis as the principal biological process upregulated specifically by HIF1A (Fig. 4c and Supplementary Fig. 7), in line with published work across many cell types[36]. In contrast, HIF2A-specific gene upregulation encompassed several GO terms, that could be grouped as pertaining to cellular structure and motility (e.g., *Igfbp5, Apela*), secretion and transport (e.g., *Fabp5, Slc3a1*), and extracellular matrix organization (e.g., *Col4a1, Npnt*) (Fig. 4d and Supplementary Fig. 8). Few genes were downregulated in a HIF1A-specific manner (Figs. 3b, 4a), and these were not enriched significantly for any GO term (Fig. 4e). However, HIF2A-specific downregulated genes were enriched significantly for assorted transmembrane transport and metabolic processes (Fig. 4e, f and Supplementary Fig. 9), that characterize the physiological function of proximal tubular cells such as re-absorption of water and solutes, suggesting an action on renal tubular differentiation.

To pursue this, we examined whether genes that were downregulated in a HIF2A-specific manner were enriched significantly for the established markers of PT differentiation that we had used to assign nephron cell type, and found a clear association ($p < 0.001$, by hypergeometric testing). To validate this further, we scored the expression of the relevant cell type-specific marker genes in each of the PT cell types and compared these across genotypes. When compared to ConKO cells, VKO and VHKO cells of PT S1, S2, and S3 cell types exhibited reduced expression of PT S1, S2, and S3 marker genes respectively (Fig. 5a), consistent with a dedifferentiation phenotype across all regions of the proximal tubule. In contrast, VEKO and VHEKO cells did not exhibit such a reduction (Fig. 5a), indicating that the phenotype was dependent specifically on HIF2A. To extend these observations we performed further experiments on selected genes using in situ RNA hybridization. This confirmed a HIF2A-dependent reduction in expression of cell-type specific marker genes for PT S1 (e.g., *Slc5a12*), PT S2 (e.g., *Inmt*), and PT S3 (e.g., *Cyp2a4*) (Fig. 5b). To provide insights as to mechanism we identified transcription factors (TFs) with binding sites near HIF2A-specific *Vhl*-dependent genes using two bioinformatic tools, LISA[37] and CheA3[38]. Five TF binding sites were ranked in the top 50 by both bioinformatic tools; three of these, *HNF4A, HNF1B*, and *FOXA2*, have well-known roles in proximal tubular or renal development[39–41] (Fig. 5c). In contrast, none of these TFs were ranked in the top 50 by either analytic tool for the set of HIF1A-specific genes (Fig. 5c), indicating that the loci of HIF2A-specific, but not HIF1A-specific genes have binding sites for TFs that influence proximal tubular differentiation.

The association between HIF2A-specific gene expression, PT-specific TFs, and dedifferentiation suggested that the HIF2A-dependent transcriptome might be sensitive to the differentiation status and tissue environment of PT cells. To assess this, we analyzed differential gene expression in VKO *versus* ConKO cells for enrichment of HIF1A- and HIF2A-specific genes that had previously been defined in

primary cultures of *Vhl*-null renal epithelial cells in vitro[12]. Gene set enrichment analysis (GSEA) showed that while these HIF1A-specific genes were significantly regulated in VKO cells of every PT identity, the HIF2A-specific genes defined in that study were not regulated in VKO cells of any PT identity (Supplementary Fig. 10).

Taken together, our analyses show that in *Vhl*-null PT cells, HIF1A and HIF2A regulate distinct biological processes, that the effects of HIF2A activation are substantially more context-dependent than HIF1A activation, and that HIF2A specifically mediates the dedifferentiation of *Vhl*-null PT cells over time in the native kidney in vivo setting.

## HIFα-isoform and time-dependent transcriptional programs in *Vhl*-null PT cells are retained in ccRCC

In previous work, we examined the evolution of *Vhl*-null dependent transcriptomic changes over time[28]. This revealed an early (within 3 weeks after *Vhl* inactivation) upregulation of genes that were dominated by those predicted to be direct HIFα targets based on reported data in other cells or tissues[42]. These changes persisted but were accompanied by additional changes that developed over time, and which encompassed up- and downregulation of genes that were not known to be direct HIFα targets. In the current work we sought to assess the HIFα-dependence of these additional time-dependent ('adaptive') changes and to test whether they were attributable to the specific activities of either HIF1A or HIF2A. To do so, we first derived lists of genes that (1) constituted this early upregulation (VKO *versus* ConKO cells at 1–3 weeks following recombination) ('Early Up'), and (2) constituted adaptive up- or downregulation specifically in *Vhl*-null cells (VKO cells at late (4–12 months)) versus early (1–3 weeks) intervals following *Vhl* inactivation ('Adaptive Up' or 'Adaptive Down'). This was performed in each PT identity separately using the same 'pseudo-bulking' methodology for differential expression analysis as used above. We then scored VHKO, VEKO, and VHEKO cells of each PT identity separately for the expression of these lists of genes. To facilitate comparisons between the genotypes, we amalgamated scores across all PT identities and scaled them such that ConKO cells had a median score of 0 for each gene list, and VKO cells had a median score of 1 or −1 for up- or down-regulated gene lists, respectively.

This analysis showed that the two HIFα isoforms had equivalent contributions to 'early' upregulation in *Vhl*-null cells (Fig. 6a). However, HIF2A inactivation had a greater effect than HIF1A inactivation in reversing 'adaptive' gene regulation in *Vhl*-null cells (Fig. 6a). Furthermore, VEKO cells exhibited similar expression scores for 'adaptive' gene lists as VHEKO and ConKO cells (Fig. 6a), highlighting the sufficiency of HIF2A inactivation in preventing adaptive gene expression in *Vhl*-null PT cells.

To test this association in the inverse manner, we scored ConKO and VKO cells harvested early or late following recombination for the expression of genes we identified above as being upregulated in a HIF1A-specific manner ('HIF1A Up'), upregulated in a HIF2A-specific manner ('HIF2A Up') and downregulated in a HIF2A-specific manner ('HIF2A Down'). Scores were calculated separately for each PT identity, amalgamated across PT identities, and scaled such that ConKO cells harvested early after recombination ('ConKO Early') had a median

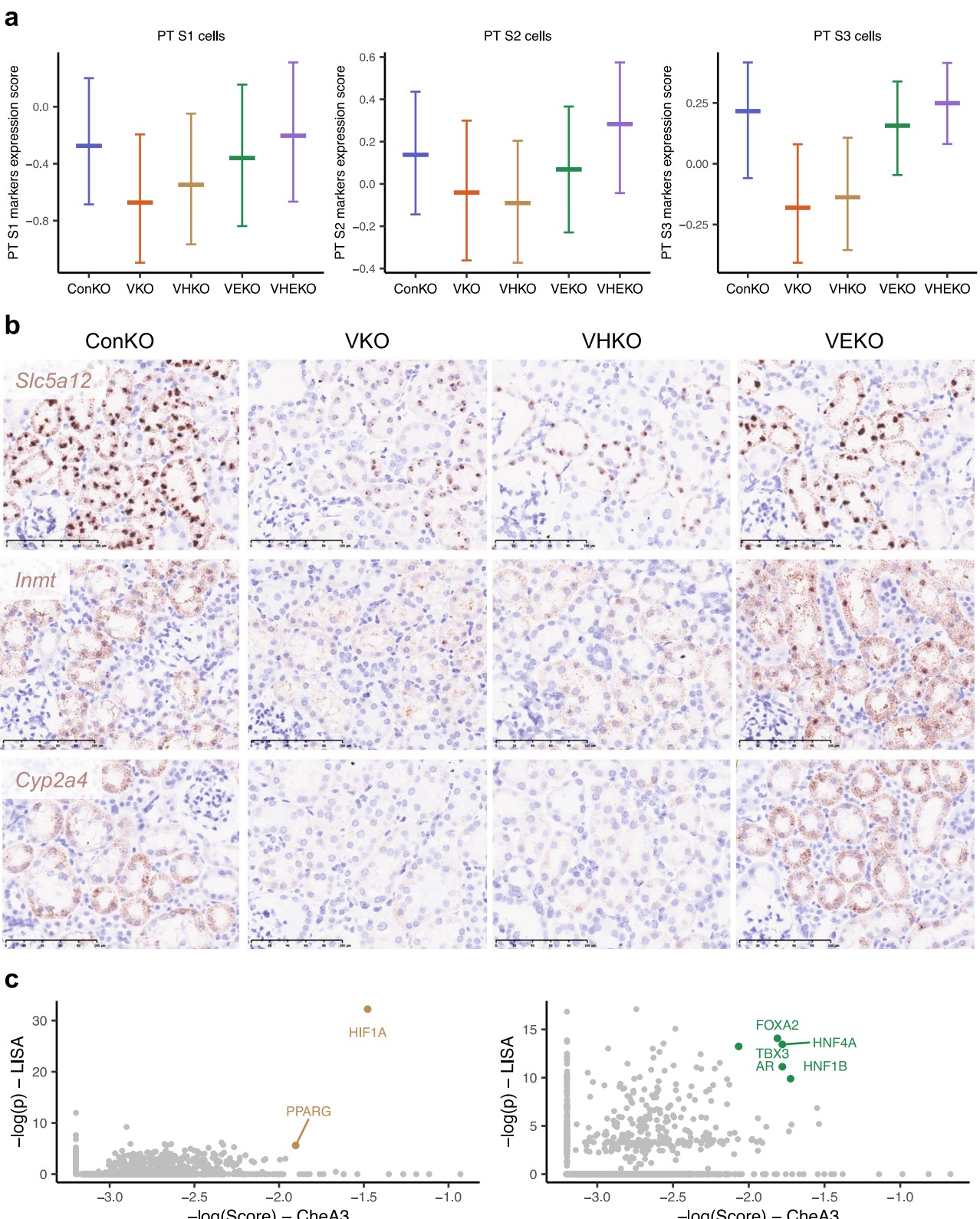

**Fig. 5 | *Vhl*-null PT cells dedifferentiate in a HIF2A-dependent manner.**
**a** Expression scores (see "Methods") for genes recognized as PT S1, S2, and S3 markers scored in PT S1, S2, and S3 cells, respectively from ConKO, VKO, VHKO, VEKO, and VHEKO mice harvested late after recombination. Data are presented as median values, with the inter-quartile range indicated by error bars.
**b** Representative RNA in situ hybridization (*n* = 3 mice per condition per transcript) depicting the expression of PT S1 marker *Slc5a12*, PT S2 marker *Inmt*, and PT S3 marker *Cyp2a4* in renal cortex of ConKO, VKO, VHKO, and VEKO mice given

5 × 2 mg tamoxifen and harvested late (4–12 months) following recombination. Scale bar denotes 100 μm; ×40 magnification. **c** Transcription factor (TF) binding sites at loci of HIF1A-specific (left) or HIF2A-specific genes (right), as predicted by LISA and CheA3. Scatter plots depict the likelihood ($\log_{10}$-transformed *p* values for LISA and a 'score' for CheA3 analysis) that binding sites for a TF are enriched at loci of the set of HIF1A-specific or HIF2A-specific genes. TFs that are ranked in the top 50 for enrichment in both LISA and CheA3 analyses are colored and labeled.

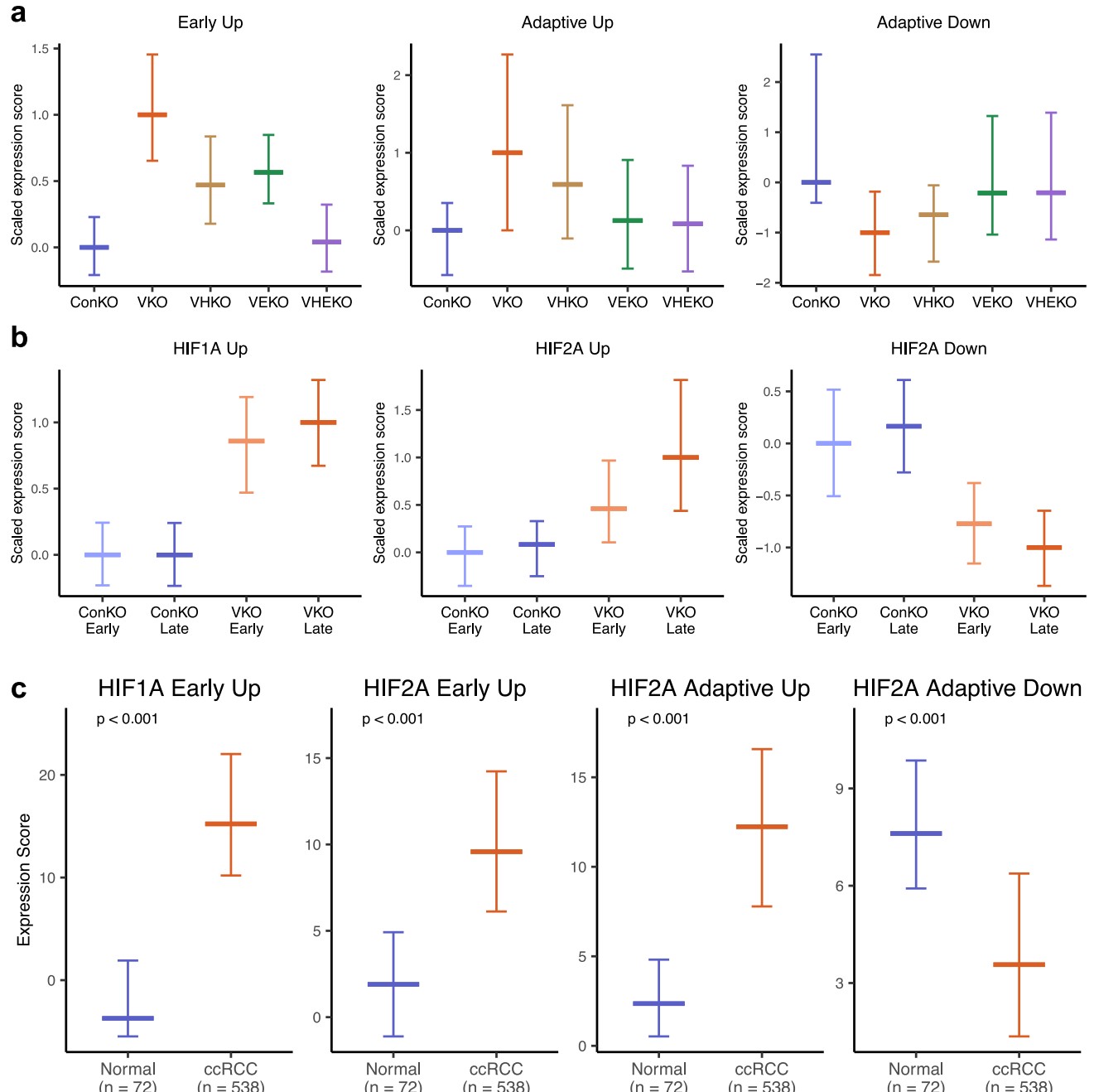

**Fig. 6 | HIF1A and HIF2A differentially contribute to early and adaptive changes in _Vhl_-null cells. a** Effect of HIFα deletion on 'early' and 'adaptive' _Vhl_-dependent gene expression. To relate the observed expression in each HIFα-inactivated genotype to the full extent of _Vhl_-dependent gene expression when HIFα isoforms are intact, the expression scores are scaled so that a score of 0 corresponds to median score for ConKO cells. The median score in VKO cells is scaled to a value of 1 for upregulated genes and −1 for downregulated genes. **b** Changes in HIF1A-specific and HIF2A-specific gene expression in _Vhl_-null cells over time. To relate the observed expression in cells to the full extent of HIFα-dependent gene expression late after _Vhl_ inactivation, the expression scores are scaled so that a score of 0 corresponds to median score for ConKO cells early after recombination. The median score in VKO cells late after recombination is scaled to

a value of 1 for upregulated genes and −1 for downregulated genes. HIF1A-dependent genes change little over time whereas greater upregulation and downregulation is observed for HIF2A-dependent genes (compare VKO Late (red) and VKO Early (orange) bars). **c** Expression of sets of genes upregulated early after _Vhl_ inactivation in a manner dependent on HIF1A ('HIF1A Early Up') or HIF2A ('HIF2A Early Up') or regulated specifically over time in _Vhl_-null PT cells in a HIF2A-dependent manner ('HIF2A Adaptive Up' and 'HIF2A Adaptive Down') in human ccRCC _versus_ normal adjacent renal tissue as available in The Cancer Genome Atlas (accessed on 14 April 2025). Pairwise comparisons by Kruskal-Wallis test with multiple testing correction using Bonferroni's method. The number of biological replicates for each condition indicated. **a–c** Data are presented as median values, with the inter-quartile range indicated by error bars.

score of 0 for each gene list and VKO cells harvested late after _Vhl_ inactivation ('VKO Late') had a median score of 1 or −1 for up- and downregulated gene lists, respectively. Comparison of scores revealed that the expression of HIF1A-specific genes reached almost its full extent early after _Vhl_ inactivation (Fig. 6b). In contrast, HIF2A-specific

genes were only partially regulated early after _Vhl_ inactivation and underwent further time-dependent changes in _Vhl_-null cells specifically (Fig. 6b).

Taken together, our transcriptomic analysis indicates that gene expression in _Vhl_-null cells is almost equally contributed to by HIF1A

and HIF2A early after *Vhl* inactivation. While HIF1A-dependent activities remain unchanged over time, HIF2A-dependent activities undergo further progressive changes that contribute to the transcriptional adaption that takes place in the expanding *Vhl*-null PT cells over time.

Finally, we asked whether and in what way, these HIFα-dependent 'early' and 'adaptive' programs were regulated in ccRCC. To do so, we first derived lists of genes which, across all PT identities, were upregulated early after *Vhl* inactivation in a HIF1A- or HIF2A-specific manner ('HIF1A Early Up' and 'HIF2A Early Up' respectively) or were up- or downregulated over time in *Vhl*-null cells in a HIF2A-specific manner ('HIF2A Adaptive Up' and 'HIF2A Adaptive Down' respectively). We then leveraged transcriptomic data in The Cancer Genome Atlas (TCGA) to compare the aggregate expression of these genes in clinical ccRCC *versus* normal kidney tissue samples. We also tested if these genes were significantly regulated in murine models of ccRCC by performing GSEA on differential expression data made available in three published studies[11,43,44]. Our analyses revealed significant regulation of each of these genes sets in all three murine ccRCC datasets (Supplementary Fig. 11) and in clinical ccRCC datasets (Fig. 6c), indicating that the HIFα isoform-specific transcriptional programs entrained early and over time following *Vhl* inactivation in PT cells in vivo are also active in advanced disease.

## Discussion

Many oncogene or tumor suppressor gene mutations exhibit marked tissue and cell specificity in their associated cancer predisposition despite causing the general dysregulation of pathways that operate widely across differentiated cells[45]. The mechanisms underlying these effects are very poorly understood even though they are likely to be of central relevance to the biology and therapy of cancer[46]. Cancer associated with inactivation of the *VHL* tumor suppressor gene provides an important example of tissue-specific oncogenesis. Biallelic inactivation of *VHL* leads to the constitutive upregulation of the HIF pathway and is strongly but almost uniquely associated with ccRCC[1,15], which is believed to be derived from the proximal RTE[4–6]. As outlined in the introduction, work on fully developed ccRCC has revealed contrasting actions of the two principal HIFα isoforms, with HIF1A being 'anti-tumorigenic' and HIF2A being 'pro-tumorigenic', though the mechanisms are unclear[9,14,16–19,21,22]. Furthermore, and somewhat surprisingly, HIF2A activating mutations are not observed in ccRCC[27,47,48], suggesting a more complex relationship between *VHL* inactivation and oncogenic HIFα activation during cancer development.

The tissue-specificity of *VHL* inactivation and HIFα activation in driving cancer might be rationalized through the study of early events in oncogenesis, when differentiation is better preserved. So far this has been hard to study, in part because of difficulty in the recognition and molecular analysis of mutated cells in native tissues, in advance of the development of identifiable morphological abnormalities. We have devised a system for oncogenic tagging that allows for the recognition and analysis of *Vhl*-null cells and have applied it to understand the role played by the two main HIFα isoforms, both of which are constitutively stabilized and transcriptionally active immediately after *Vhl* inactivation. To do this, we co-deleted *Hif1a*, *Epas1*, or both *Hif1a* and *Epas1*, together with *Vhl* using tamoxifen-inducible expression of Cre recombinase in the RTE. In tagged *Vhl*-null cells, very high levels of co-deletion of the genes encoding HIFα isoforms were observed. Whilst not every single tagged cell harbored genetic co-deletion of HIFα (and we therefore may have underestimated the effects of HIFα co-deletion), our strategy revealed distinct roles for the two HIFα isoforms that are of relevance to early oncogenesis following *Vhl* inactivation.

Our analysis suggested that the tissue-specificity of *VHL*-associated oncogenesis is derived from the isoform-specific and cell type-specific actions of the two HIFα isoforms. Inducing *Vhl* inactivation in papillary RTE cells resulted in their loss from the renal papilla, possibly explaining why *VHL*-associated cancers do not appear to arise from

this site. This cell loss was rescued completely and specifically by *Hif1a* co-deletion, marking HIF1A as potentially 'anti-tumorigenic' and HIF2A as agnostic to tumorigenesis in cells in this region of the kidney. Large-scale genetic studies in many types of cultured cells have identified *VHL* inactivation and HIF1A activation to have negative effects on cell survival (https://depmap.org/portal/gene/VHL)[49–51]. Our analysis in vivo has extended these observations to papillary epithelial cells in the normal kidney.

In this context, our observations regarding HIFα activation in the cancer-associated PT cells in the renal cortex and outer medulla are of particular interest. Here, inducing *Vhl* inactivation did not result in cell loss but rather resulted in a significant accrual of tagged *Vhl*-null PT cells. That this represented cell proliferation was supported by increased numbers of tdTomato-positive cells co-expressing the marker of proliferating cells Ki67 and by increased contiguous grouping of tdTomato-positive cells in tissue. Notably, the co-inactivation of either of the two HIFα isoforms impaired all these measures of proliferation, indicating that in this setting both HIFα isoforms are tolerated and that the activation of each is necessary for the initial proliferation and growth of cells in the proximal tubule. This requirement for both HIFα isoforms potentially contrasts with work in mouse models in which *Hif1a* co-deletion with *Vhl* failed to prevent the formation of *Vhl*-null renal cysts[26], or conversely in which, stable overexpression of either HIF1A[52] or HIF2A[8] alone induced proliferative lesions in the RTE. However, it is consistent with work in mouse models of ccRCC, in which the onset of renal tumors following concomitant *Vhl, Trp53*, and/or *Rb1* inactivation was dependent on the integrity of both HIF1A and HIF2A[11,12]. Unlike these models, our analysis of *Vhl*-dependent proliferation has been performed with single-cell resolution and in advance of the development of any morphological abnormalities in the kidney. We have thus been able to identify that combined stabilization of HIF1A and HIF2A is a primary driver of cell proliferation in PT cells.

A dependency on both HIFα isoforms may arise either because the two HIFα isoforms independently regulate a shared set of genes critical for *Vhl*-null cell proliferation, or because the two HIFα isoforms regulate distinct sets of genes that act in concert to confer a proliferative advantage to *Vhl*-null cells. Our single-cell transcriptomic analysis indicated the latter to be most likely in *Vhl*-null cells. This is because in cells of each region of the PT, the sets of genes regulated individually by HIF1A and HIF2A showed little overlap and were enriched for distinct sets of biological pathways. While HIF1A-regulation was clearly linked to early upregulation of glycolytic genes, HIF2A-mediated gene regulation encompassed a range of pathways including lipid and amino acid metabolism, extracellular matrix reorganization, and dedifferentiation. Concomitant activation of these processes may allow *Vhl*-null cells to adopt several 'hallmarks' of cancer such as 'deregulated metabolism', 'phenotypic plasticity', and non-homeostatic invasion and growth, and thus aid oncogenesis[53,54]. Our observations of largely discrete effects of *Hif1a* and *Epas1* inactivation also provide one explanation for the paradox that *VHL* inactivating mutations are almost always observed in ccRCC, but *EPAS1* activating mutations are not[27,47,48]. Our finding of early co-dependence on HIF1A and HIF2A suggests that at this stage, the inability of activated HIF2A to regulate the repertoire of HIF1A-specific genes including metabolic targets, might make *EPAS1* activating mutations insufficient to confer a proliferative advantage to *Vhl*-null cells.

In addition to cell-type specificity, we also observed distinct time-dependence of HIF1A- and HIF2A-dependent gene regulation. In previous work, we studied how gene expression changed both early and late after *Vhl* inactivation. That work demonstrated two major effects; a change in gene expression that occurred early after *Vhl* inactivation and persisted, and an additional set of changes in gene expression that occurred only after time and included downregulation of markers of PT differentiation[28]. Our analysis here revealed that while the changes in gene expression observed early following *Vhl* inactivation were

contributed to equally by both HIFα isoforms, the delayed changes that were super-imposed on these, including dedifferentiation of PT cells, were largely dependent on HIF2A. Thus, ultimately HIF2A contributed substantially more than HIF1A to altered gene expression in *Vhl*-null PT cells. It is unclear why the effects of HIF2A activation specifically take so long to manifest fully, though it is possible that new HIF2A targets are made accessible in chromatinized DNA as *Vhl*-null cells dedifferentiate.

Finally, we found that the HIF1A- and HIF2A-specific early and adaptive transcriptional programs identified in *Vhl*-null PT cells were retained in ccRCC. It is unclear which aspects of these programs are necessary for, or which only associate with oncogenesis. In either case, the relevance of these programs to ccRCC, and the substantial dependency on HIF2A of gene expression and proliferation of *Vhl*-null cells in the proximal tubule provides support for the early use of HIF2A antagonism (e.g., with the clinically approved inhibitor Belzutifan) to prevent the expansion of potential cancer cells-of-origin early in VHL disease.

Overall, our data demonstrates that the major consequences of *Vhl* inactivation in vivo – cell-type specific elimination, proliferation, and gene expression – are all dependent on the cell-type specific, temporally-distinct and isoform-specific activities of HIFα. Additionally, we provide evidence that pro-oncogenic effects manifest only when *Vhl* inactivation entrains the disparate activities of both HIFα isoforms in concert. Further work will be required to dissect the role of delayed HIF2A-dependent gene regulation in *Vhl*-null cells and test the effects of early HIF2A inhibition in preventing *Vhl*-null PT cells from proliferating in tissue.

## Methods

### Mice

All experimental procedures were conducted following approval by the Medical Science Ethical Review Committee of the University of Oxford and authorized under UK Home Office regulations of Animals (Scientific Procedures) Act 1986.

Mice were housed in individually ventilated cages on a 13 h light/ 11 h dark cycle with food and water provided *ad libitum*. B6.*Vhl*$^{tm1.1b(tdTomato)Pjr}$ (*Vhl*$^{pjr,fl}$) mice were commissioned from Ozgene, Australia, and generated using goGermline technology[55]. *Vhl*$^{tm1jae}$ mice[56] (*Vhl*$^{jae,fl}$) (RRID: IMSR_JAX:012933) were crossed with *Tg(Pgk1-cre)1Lni*[57] (*Pgk1-Cre*) (RRID: IMSR_JAX:020811) to generate a constitutively inactivated *Vhl*$^{tm1jae}$ allele (*Vhl*$^{jae,KO}$). *Tg(Pax8-cre/ERT2)CAmat* (*Pax8-CreERT2*) (RRID: IMSR_HAR:9175) mice were obtained via EMMA[29]. Mice carrying conditional alleles for *Hif1a* (*Hif1a*$^{tm3Rsjo}$; RRID: MGI:6863863; termed *Hif1a*$^{fl}$)[58] and *Epas1* (*Epas1*$^{tm1.1Mcs}$; RRID:MGI:3710345; termed *Epas1*$^{fl}$)[59] were gifts from Randall Johnson and M Celeste Simon respectively. These lines were generated on a B6.129 mixed background but were inter-crossed with C57B6/J animals for at least five generations prior to starting experiments.

*Vhl*$^{wt/pjr,fl}$; *Pax8-CreERT2* ("ConKO"), *Vhl*$^{jae,KO/pjr,fl}$; *Pax8-CreERT2* ("VKO"), *Vhl*$^{wt/pjr,fl}$; *Hif1a*$^{fl/fl}$; *Pax8-CreERT2* ("ConHKO"), *Vhl*$^{jae,KO/pjr,fl}$; *Hif1a*$^{fl/fl}$; *Pax8-CreERT2* ("VHKO"), *Vhl*$^{wt/pjr,fl}$; *Epas1*$^{fl/fl}$; *Pax8-CreERT2* ("ConEKO"), *Vhl*$^{jae,KO/pjr,fl}$; *Epas1*$^{fl/fl}$; *Pax8-CreERT2* ("VEKO"), *Vhl*$^{wt/pjr,fl}$; *Hif1a*$^{fl/fl}$; *Epas1*$^{fl/fl}$; *Pax8-CreERT2* ("ConHEKO"), and *Vhl*$^{jae,KO/pjr,fl}$; *Hif1a*$^{fl/fl}$; *Epas1*$^{fl/fl}$; *Pax8-CreERT2* ("VHEKO") mice of both sexes were administered tamoxifen (2 mg dose daily for five consecutive days) when they were >20 g in body weight. Mice were harvested either 1–3 weeks (early) or 4–12 months (late) after recombination was induced.

### Genotyping and PCR

Genotypes of all experimental mice were confirmed at time of harvest. Primer sequences and expected product sizes were as follows: *Vhl*$^{jae,KO}$: 5′-CTGGTACCCACGAAACTGTC-3′, 5′-CTAGGCACCGAGCTTA GAGGTTTGCG-3′ and 5′-CTGACTTCCACTGATGCTTGTCACAG-3′ (260 bp for *Vhl*$^{wt}$, 260 and 240 bp for *Vhl*$^{jae,KO}$ multiplex products);

*Vhl*$^{pjr,fl}$: 5′-GGTGCTAATTGAAGGAAGCTACTG-3′ and 5′-CTCCTCCGAG GACAACAACATG-3′ (1067 bp); *Pax8-CreERT2*: 5′-CGGTCGATGCAAC GAGTGATGAGG-3′, 5′-CCAGAGACGGAAATCCATCGCTCG-3′, 5′-CTCA TACCAGTTCGCACAGGCGGC-3′ and 5′-CCGCTAGCACTCACGTTGG TAGGC-3′ (300 bp and 600 bp multiplex products); *Hif1a*$^{fl}$: 5′-TTGGGGGATGAAAACATCTGC-3′, 5′- GCAGTTAAGAGCACTAGTTG-3′, 5′- GGAGCTATCTCTCTAGACC-3′ (240 bp for *Hif1a*$^{wt}$, 260 bp for *Hif1a*$^{fl}$, 270 bp for *Hif1a*$^{KO}$); *Epas1*$^{fl}$: 5′- CAGGCAGTATGCCTGGCT AATTCCAGTT-3′, 5′- CTTCTTCCATCATCTGGGATCTGGGACT-3′, and 5′- GCTAACACTGTACTGTCTGAAAGAGTAGC-3′ (360 bp for *Epas1*$^{KO}$, 430 bp for *Epas1*$^{wt}$, 460 bp for *Epas1*$^{fl}$).

### Immunohistochemistry (IHC)

**Tissue preparation.** Mice were euthanized with terminal isoflurane anesthesia, flushed via the aorta with 1x PBS pH 7.4 (Gibco 70011) to clear blood from tissues, and perfused with 4% (w/v) paraformaldehyde (Sigma P1213) in PBS at room temperature (RT). Kidneys were bisected and fixed in 10% neutral buffered formalin (NBF; Sigma HT501128) with rocking at RT for 24 h. Kidneys were then dehydrated through a graded ethanol series (70% to 100%) and xylene before paraffin embedding. Embedded tissues were cut to 4 μm sections on a Thermo Microm HM 355S Microtome using MB35 Premier Blades. Sections were floated on warm distilled water and mounted on poly-lysine coated slides (Fisher 10149870). Slides were dried for at least 3 h at 37 °C before IHC.

**tdTomato IHC.** Sections were deparaffinized with xylene and ethanol, rehydrated with double-distilled water, and subjected to heat-induced epitope retrieval (HIER) with TE buffer (10 mM Tris, 1 mM EDTA, pH 9.0) in a steamer (at 95 °C) for 20 min. Endogenous peroxidase activity was blocked by incubation with Dako Peroxidase Blocking solution (Agilent S2023) for 10 min at RT. Non-specific protein binding was blocked by incubation with 5% (w/v) bovine serum albumin (BSA; Sigma 5482) in 1x TBS-T (50 mM Tris, 31.6 mM NaCl, 0.1% (v/v) Tween-20; pH 8.4) for 40 min at RT. tdTomato was detected by overnight incubation at 4 °C with an α-RFP antibody (Rockland 600-401-379, RRID:AB_2209751) diluted 1:1000 in Dako Antibody Diluent Solution (Agilent S3022). Slides were washed in 1x TBS-T. Signal was detected with the Dako Envision system (Agilent K4003), visualized using diaminobenzidine (DAB), and counterstained with modified Harris Hematoxylin (Fisher 72711) differentiated for 10 seconds in 0.25% (w/v) HCl in 70% ethanol. Hematoxylin staining was blued by immersion in 0.06% (w/v) NH$_4$OH in water for 30 seconds. Slides were dehydrated and mounted with DPX mountant (Merck 06522).

**tdTomato-Ki67 dual IHC.** Slides were deparaffinized and rehydrated as above and were subjected to HIER in Dako Antigen Retrieval Solution (Agilent S1699) in a pressure cooker (at 120 °C) for 12 min. Endogenous peroxidase activity was blocked by incubating slides in 3% (v/v) H$_2$O$_2$ (Merck H1009). Endogenous avidin and biotin were then blocked with the Abcam Avidin/Biotin blocking kit (Abcam ab64212). Non-specific protein binding was then blocked with a serum-free protein block solution (Agilent X090930-2) and 5% BSA in 1x TBS-T. Ki67 was detected using a biotinylated Rabbit anti-Ki67 biotin-conjugated antibody (Life Technologies 13569882, RRID: AB_2572794; 1:200) at RT for 2 h. Signal was generated by incubation with streptavidin-HRP (BD Biosciences 550946) at RT for 30 min and visualized using DAB (brown). Peroxidase activity on these sections was then neutralized by incubation with Dako Peroxidase Blocking solution (Agilent S2023) for 10 min at RT before starting a second round of IHC for tdTomato. A complete second round of HIER, blocking, and antibody incubations were performed as above. tdTomato signal was visualized using the Vector VIP (purple) substrate (Vector SK4605), which contrasted with the DAB (brown) signal produced previously on the same sections for Ki67. Sections were counterstained by incubation in Methyl green

(Vector H3402500) for 15 min at 60 °C, dehydrated through quick rinses in 100% ethanol and xylene, and mounted in DPX mountant.

**Microscopy.** IHC sections were scanned with a Hamamatsu Nano-Zoomer S210 slide scanner at 40x magnification and analyzed with Hamamatsu NDP.view2 software.

**Quantification.** The proportion of cells positive for tdTomato or Ki67 by IHC was quantified using the HALO Image Analysis Software (v3.5 and v3.6; Indica Lab, Albuquerque, USA). Kidney sections were annotated manually to define regions of the renal papilla or the renal cortex and outer medulla based on morphological appearance and anatomical location in the tissue[60]. Briefly, the renal cortex was identified by the presence of glomeruli and of proximal convoluted tubules (corresponding to PT S1 and S2 cell types), distal convoluted tubules, and cortical collecting ducts, each of which have characteristic tubular shape, nuclear shape, and nuclear distribution along the tubular circumference. The outer medulla was distinguished from the cortex by the absence of glomeruli and the presence of proximal straight tubules (corresponding to PT S3 cell type); it was distinguished from the inner medulla by the absence of cells of the loop of Henle, which are thinner and more densely packed than cells of the proximal straight tubule. The renal papilla, comprising cells of the collecting duct and located at the apex of the medullary pyramid, was characterized by its elongated tubules and sparse cellularity, giving it a relatively pale appearance compared to the cortex and outer medulla. Nuclei were detected using the HALO AI v3.6 Nuclei Seq algorithm based on hematoxylin (for tdTomato IHC) or methyl green (for tdTomato-Ki67 dual IHC) counterstaining. Nuclei <15 µm$^2$ and >200 µm$^2$ were excluded from analysis. The Red-Green-Blue (RGB) IHC image was deconvoluted computationally into individual contributions from DAB, VIP, hematoxylin, or methyl green based on known colorimetric properties of the three stains. Cells were recorded as tdTomato-positive if the tdTomato (DAB for tdTomato IHC, VIP for tdTomato-Ki67 dual IHC) signal passed a fixed intensity threshold in detected nuclei and in a 2 µm perimeter drawn around each nucleus. Cells were scored as positive for Ki67 if the signal (DAB in tdTomato-Ki67 dual IHC) passed a fixed intensity threshold in the nucleus. For tdTomato IHC, an average of 185,189 cells, and 16,109 cells per mouse were quantified in the cortex + outer medulla and papilla, respectively. For tdTomato Ki67 dual IHC, an average of 125,384 cells per mouse were quantified in the cortex and outer medulla.

### RNA in situ hybridization
RNA in situ hybridization was performed using either RNAscope 2.5 HD Assay – BROWN (ACD BioSciences 322300) for single assays, or RNAscope 2.5 HD Duplex Detection Kit (ACD BioSciences 322430) for dual assays, on 4 µm paraffin sections cut on the previous day and baked at 60 °C for 1 h. Probes used were: *Pgk1* (Cat. 312961), *Angptl3* (Cat. 576601), *Slc5a12* (Cat. 839881), *Inmt* (Cat. 486371), *Cyp2a4* (Cat. 884531), *Neat1* (Cat. 440351) and *Fxyd2* (Cat. 572631-C2). For single assays, sections were counterstained with hematoxylin as described for IHC. For dual assays, sections were not counterstained.

### Tissue dissociation
Kidneys were bisected, the renal capsule removed, and then macerated for up to 7 min on a bed of ice. Macerated kidneys were then subjected to single-cell dissociation using the Multi-Tissue Dissociation Kit 2 (Miltenyi 130-110-203). Briefly, macerated tissues were suspended in 1.45 ml Buffer X, 30 µl of Enzyme D, 15 µl of Enzyme P, 15 µl of Buffer Y, and 6 µl of Enzyme A, all prepared according to the manufacturer's instructions. The suspension was then incubated under water at 37 °C for 30 min in a shaking incubator set to 150 rpm. The procedure was stopped by the addition of 150 µl fetal bovine serum (FBS; Sigma F7524) and resuspension in 9 ml of RPMI-1640 medium (Merck R0883). The digest was filtered through a 40 µm cell strainer to remove undigested tissue and the filtrate was centrifuged (300 g for 10 min at 4 °C). Erythrocytes were eliminated by resuspending dissociated cells in 3 ml of 1x RBC Lysis Buffer (Miltenyi 130-094-183) prepared in deionized water and incubating for 2 min at RT. Cells were centrifuged (300 × g for 5 min at 4 °C) and resuspended in ice-cold D-PBS before being counted on a Thermo Scientific Countess II machine for total yield and viability.

### Fluorescence activated cell sorting
Dissociated cells were resuspended in 10% FBS, 2 mM EDTA, in D-PBS (Thermo 14190144) to a concentration of 5,000,000 live cells per ml. DAPI (Sigma D9542) was added to stain for viability. Cells were sorted using a BD Aria Fusion Cell Sorter. tdTomato was excited with a 561 nm laser and fluorescence detected with a 582/15 band pass filter. DAPI was excited with a 405 nm laser and fluorescence detected with a 450/40 band pass filter. The gating strategy is provided in Supplementary Fig. 12. Live, single, tdTomato-positive cells were collected in FBS-coated polypropylene tubes and pelleted by centrifugation at 300 g for 10 min. Cells were counted again for yield and viability and processed for DNA extraction or scRNA-seq.

### Single-cell library preparation and sequencing
Sorted cells were prepared into single-cell droplets using the Chromium Next GEM Single-cell kit. 20,000 live cells per sample were loaded on separate Chromium Next GEM Chip G (10X Genomics PN-1000127) channels. cDNA clean-up, amplification and adaptor ligation were performed with the Chromium Next GEM Single Cell 3′ Kit v3.1 (10X Genomics PN-1000268). cDNA yield was quantified using the High Sensitivity D1000 ScreenTape Assay (Agilent 5067-5584 and 5067-5585) to optimize the number of reaction cycles for library preparation. Single cell sequencing libraries were prepared using the Library Construction Kit (10X PN-1000190) and Dual Index Kit TT Set A (10X Genomics PN-1000215) sequencing indices.

Single-cell libraries were sequenced on an Illumina NextSeq 2000 sequencing system with P3 200 cycle reagents (Illumina 20040560). Libraries from up to six samples were pooled in an equimolar ratio, diluted to 650 pM in RSB buffer and mixed with 1% PhiX Control v3 DNA (Illumina FC-110-3001) for sequencing. Preliminary sequencing results (bcl files) and FASTQ files were generated with the DRAGEN FASTQ Generation, or the DRAGEN BCL Convert workflow optimized for Single Cell RNA Library Kit 1 library prep kit and the Single Cell RNA Index Adaptors 1-B index adaptor kit, with 28 and 152 Read 1 and 2 cycles, respectively.

### Single-cell sequencing data processing and quality control
The mm10 reference genome was downloaded from https://cf.10xgenomics.com/supp/cell-exp/refdata-gex-mm10-2020-A.tar.gz. The transcript sequence and annotation for $Vhl^{pjr-KO}$, that included the tdTomato transgene, were added manually to the FASTA reference genome file and GTF file respectively using the CellRanger (v6.1.1) mkref function. FASTQ files generated for each sample were aligned to the custom reference genome by CellRanger using the default parameters. After aligning, for each read pair, cell barcodes and unique molecular identifiers (UMIs) were obtained from Read 1 and read counts per feature were obtained from Read 2. Only those UMIs that could be linked to a valid cell barcode and a gene exon region were included to create the cell by gene count matrix. Reads from *Vhl* exon 1 were excluded from analysis to prevent ambiguous alignment to $Vhl^{wt}$ or $Vhl^{pjr-KO}$ alleles. Cells were subjected to the following filters: detected genes per cell >200, fraction of mitochondrially-encoded reads per cell <0.5, and detected genes <3*median for each sample. The threshold for mitochondrially encoded reads was set to this value in line with published kidney scRNA-seq studies to account for the high mitochondrial content in the RTE cells[30]. Data from cells from different samples and sequencing runs were then aggregated, which sometimes

required manual renaming of duplicate cell 'names'. Downstream analysis was conducted using the R package Seurat v4.0.3[61] and v5.1.0[62].

## Dimension reduction and clustering

Dimension reduction was performed either on all sequenced cells together, or on cells of each PT identity separately. In each case, sex-specific gene expression was considered by performing 'batch correction' based on the sex of the mouse from which cells were derived. Briefly, for cells of each sex, read counts were log-normalized and the top 2,000 varying features were identified and analyzed using the FindVariableFeatures function of Seurat. Next, features were selected for downstream integration using the SelectIntegrationFeatures function. Normalized read counts for these features were scaled before performing principal component analysis (PCA). Following this, anchoring features between the two sexes were calculated using FindIntegrationAnchors functions with parameter "reduction='rpca'", which used reciprocal PCA (top 30 PCs) to identify an effective space in which to find anchors. Finally, samples of the two sexes were integrated with the IntegrateData function using the identified anchoring features.

After integration, expression of these genes was scaled to have a mean at 0 and standard deviation of 1 using the ScaleData function. PCs were analyzed for the expression of these genes using the RunPCA function. The first 30 components were then included for shared nearest neighbor modularity optimization with the Louvain algorithm using the FindClusters function and resolution set to 0.2. Finally, UMAP dimension reduction was performed using the RunUMAP function with default parameters and 30 dimensions.

## Cell type assignment

Cell type specific marker genes were obtained from published scRNA-seq studies of kidneys from wild-type C57BL/6 J mice[31–34]. For each cell type described in these papers, marker genes were selected as those with $\log_2$-fold change (compared to other cells in the dataset) >1.0, and those that were expressed in >50% of cells they marked, and <20% of the cells they did not mark. The resultant list was cross-referenced to a published meta-analysis of renal cell type assignment performed across eight different mouse kidney scRNA-seq datasets[35]. PT S2 and S3 cells have been shown to exhibit sex-specific gene expression[63]. Accordingly, sex-specific lists of marker genes were prepared for these cell types[34]. The four papers to which we referred did not always divide renal cell types in the same manner. For our analysis, conflicts in renal cell type classification were resolved by reference to histological and anatomical literature on the kidney[64,65]. Altogether, marker gene lists were prepared for the following cell types: three segments of the proximal tubule PT S1, S2, and S3; Loop of Henle epithelium, Distal convoluted tubule, principal and intercalated cells of the collecting duct, parietal cells, podocytes, pericytes, endothelial cells, fibroblasts, macrophages, NK cells, monocytes, neutrophils, B lymphocytes, and T lymphocytes. These lists are provided in Supplementary Data 3.

The collective expression of each set of cell-type specific marker genes was then scored in each cell using the 'AddModuleScore_UCell' function from the UCell package (v2.8.0)[66], with the 'maxRank' parameter set to 1500. This scoring method was used as it allows scores for different sets of genes to be compared within the same cell[66]. Cell type was finally assigned for each individual cell by identifying the set of cell-type specific marker genes that attained the highest expression score in that cell.

## PT class assignment

Sets of genes exhibiting anti-correlated expression within PT S1, S2, and S3 cells, termed 'PT Module A' and 'PT Module B', have been described previously[28] and are provided in Supplementary Data 1. PT cells were scored for the expression of these sets of genes using the 'AddModuleScore' function[67] in Seurat with 100 bins and 50 controls per bin. This scoring method was used as it allows for the expression of the same set of genes to be compared across cells in a dataset. PT Class A cells were identified as those with PT Module A expression score >0.125 and the others were identified as PT Class B cells.

## Differential expression analysis

To account for the known heterogeneity in baseline and *Vhl*-dependent gene expression across different types of renal cells, we performed differential expression analysis separately in subsets of cells grouped according to their PT identity. This analysis was performed using a 'pseudo-bulking' approach, that treats each mouse, and not each cell, as a biological replicate, and thereby reduces the extent of false discoveries that can be made in scRNA-seq analysis[68,69]. Raw gene expression counts in cells from each mouse were first summed using Seurat's 'AggregateExpression' function to generate a 'sample × gene' matrix from a 'cell × gene' matrix. The summed data was then used to calculate differential expression using the 'DESeqDataSetFromMatrix' function from the DESeq2 package (v1.44.0)[70], with the formula set to "~ sex + genotype" to account for sex-dependent differences in baseline gene expression across mice. Genes were considered differentially expressed if they were robustly altered ($|\log_2$-fold change (L2FC)$| > 1.0$), and if the alteration was statistically significant (adjusted $p$ value < 0.05).

**HIFα-dependent gene expression.** The following comparisons were made: VKO *versus* ConKO, VHKO *versus* VKO, VEKO *versus* VKO, and VHEKO *versus* VKO mice, all harvested late after recombination. Genes that were robustly and significantly upregulated or downregulated in VKO *versus* ConKO mice were deemed to be '*Vhl*-dependent genes'. HIFα-dependence of *Vhl*-dependent genes was determined by the following:

- If *Vhl*-dependent genes were regulated significantly in the opposite direction in VHKO *versus* VKO mice than in VKO *versus* ConKO mice, and if the magnitude of L2FC in VHKO *versus* VKO mice was more than half of the magnitude of L2FC in VKO *versus* ConKO mice, they were called '*Hif1a*-dependent genes'.
- If *Vhl*-dependent genes were regulated significantly in the opposite direction in VEKO *versus* VKO mice than in VKO *versus* ConKO mice, and if the magnitude of L2FC in VEKO *versus* VKO mice was more than half of the magnitude of L2FC in VKO *versus* ConKO mice, they were called '*Epas1*-dependent genes'.
- If *Vhl*-dependent genes were regulated significantly in the opposite direction in VHEKO *versus* VKO mice than in VKO *versus* ConKO mice, and if the magnitude of L2FC in VHEKO *versus* VKO mice was more than half of the magnitude of L2FC in VKO *versus* ConKO mice, they were called '*Hif1a/Epas1*-dependent genes'.

Genes that were '*Hif1a*-dependent', '*Epas1*-dependent', or '*Hif1a/Epas1*-dependent' were called 'HIFα-dependent genes'. Genes that did not pass these thresholds but were '*Vhl*-dependent' were deemed to have 'Ambiguous HIF dependence'.

Isoform specificity of HIFα-dependent genes was determined by the following:

- Genes that were neither '*Hif1a*-dependent' nor '*Epas1*-dependent' were identified as requiring inactivation of both isoforms for their regulation ('HIF1A + HIF2A').
- Genes that were '*Hif1a*-dependent' but not '*Epas1*-dependent' were identified as requiring HIF1A alone for their regulation ('HIF1A alone or 'HIF1A-specific').
- Genes that were '*Epas1*-dependent' but not '*Hif1a*-dependent' were identified as requiring HIF2A alone for their regulation ('HIF2A alone or 'HIF2A-specific').
- Genes that were '*Hif1a*-dependent' and '*Epas1*-dependent' were identified as requiring inactivation of either isoform for their regulation ('HIF1A or HIF2A').

**Early and adaptive gene expression**. To distinguish early and adaptive changes, the following comparisons were made: VKO *versus* ConKO mice harvested early (1–3 weeks) after recombination, VKO mice harvested late (4–12 months) *versus* early after recombination, and ConKO mice harvested late versus early after recombination. Genes that were robustly and significantly upregulated or downregulated in VKO *versus* ConKO mice harvested early after recombination were identified as part of 'early' gene regulation following *Vhl* inactivation. Adaptive changes were identified by evaluating differential gene expression at late *versus* early intervals following recombination in VKO and ConKO mice separately. Genes regulated in a time-dependent manner in VKO mice but not in ConKO mice were identified as part of 'adaptive' gene regulation following *Vhl* inactivation.

## Over-representation analysis

Over-representation of biological processes GO terms within sets of genes that are "HIF1A-specific" or 'HIF2A-specific' for their *Vhl*-dependent up- or downregulation was evaluated using the enrichGO function of the clusterProfiler package (v4.12.6)[71], with multiple testing corrected using false discovery rate. GO terms with recommended restrictions over their use for direct gene product annotation, and those with fewer than 50 genes and more than 1000 genes were excluded from analysis. To provide a guide as to the extent of dysregulation among over-represented genes, HIF1A-specific and HIF2A-specific genes that were members of each significantly over-represented GO term (adjusted $p < 0.01$) were identified. L2FC values for each of these genes in VHKO *versus* VKO and VEKO *versus* VKO were then calculated across all PT identities in which the genes were regulated by *Vhl* inactivation. The L2FC values for member genes within each GO term were then averaged to provide a measure of the magnitude of change. Significantly over-represented GO terms were then ordered by hierarchical clustering based on these average L2FC values using the hclust function in R, with method set to 'ward.D2' and plotted as a heatmap.

Cell-type specific marker genes for PT S1, PT S2 male, PT S2 female, PT S3 male, and PT S3 female cells, used previously to assign cell type, were pooled together to define a set of 'PT marker' genes. Over-representation of these genes within the set of HIF2A-specific downregulated genes was then evaluated by hypergeometric testing using the phyper function in R.

## Gene set enrichment analysis (GSEA)

GSEA analyses and plotting were performed with the 'fgsea' and 'plotEnrichment' functions of the fgsea[72] package (v1.30.0) in R, using gene sets and ranked lists of genes as follows.

To evaluate the enrichment in *Vhl*-null cells of each PT identity in vivo of HIFα-dependent genes as defined in primary cultures of *Vhl*-null renal epithelial cells, genes were ranked by their L2FC values in VKO *versus* ConKO cells at the late timepoint as described above. Lists of HIF1A-specific and HIF2A-specific genes in primary cultures of *Vhl*-null renal epithelial cells were obtained from supplementary material provided by Schonenberger et al.[12].

To evaluate the enrichment of internally-identified HIF1A- and HIF2A-specific genes that were regulated early or over time in *Vhl*-null PT cells in experimental mouse ccRCC samples, data on differential expression in ccRCC tumor *versus* normal cortical samples was obtained from the supplementary material of three published reports of murine models of ccRCC: (1) *Vhl/Rb1/TrpS3*-null ccRCC *versus* wild-type cortex samples from Harlander et al.[43], (2) *Vhl/Rb1/TrpS3*-null ccRCC *versus* wild-type cortex samples from and Hoefflin et al.[11], and (3) *Vhl/Pbrm1*-null ccRCC *versus* wild-type cortex samples from Nargund et al.[44]. Gene names were converted using the 'orthologs' function of the babelgene[73] package (v22.9) in R. Genes were ranked by their L2FC values in tumor *versus* cortical samples in each dataset. Genes which across any PT identity were upregulated in VKO *versus*

ConKO cells at the early timepoint, and which were identified to be 'HIF1A-specific' or 'HIF2A-specific' were added to the 'HIF1A Early Up' and 'HIF2A Early Up' gene sets respectively. Genes which across any PT identity were up- or downregulated at the late *versus* early timepoint specifically in VKO cells, and which were identified to be 'HIF2A-specific' were added to the 'HIF2A Adaptive Up' and 'HIF2A Adaptive Down' gene sets respectively.

## LISA and CheA3 analysis

Transcription factors (TFs) predicted to have binding sites on the repertoire of genes that were 'HIF1A-specific or 'HIF2A-specific were predicted using two analytical tools—LISA[37] (http://lisa.cistrome.org/) and CheA3[38] (https://maayanlab.cloud/chea3/)—using default parameters for both tools. Consensus TFs were identified as those that were ranked in the top 50 by both tools for each set of genes. TFs were plotted according to the '1st sample $p$ value' for LISA analysis and according to the 'Score' for CheA3 analysis.

## Scoring gene set expression

The collective expression of defined sets of genes in scRNA-seq data was evaluated using the 'AddModuleScore' function[67] of Seurat, with 100 bins and 50 controls per bin. This method was chosen so that expression of the same set of genes can be compared across cells in the dataset[67].

For scoring PT marker gene expression, PT S1, S2, and S3 cells were first separated into subsets. PT S2 and PT S3 cells, known to exhibit sexually dimorphic gene expression[34,63], were further separated into male and female subsets. These subsets were then scored for the expression of PT S1, PT S2 male, PT S2 female, PT S3 male, or PT S3 female cell-type specific marker genes as appropriate.

Expression of genes that are part of early and adaptive regulation following *Vhl* inactivation was scored in a PT identity dependent manner. First, genes that were regulated early after *Vhl* inactivation ('Early Up') and that were part of an adaptive regulation specifically in *Vhl*-null cells ('Adaptive Up' and 'Adaptive Down') were defined for each PT identity. Genes that were identified as part of both the early and adaptive changes were excluded from this analysis. Then, cells of each PT identity were separated into subsets and scored for the corresponding set of 'Early Up', 'Adaptive Up' and 'Adaptive Down' genes. Genotype-dependent differences in early and adaptive regulation were then visualized for cells of all PT identities plotted together.

HIFα isoform specific gene expression was scored in a PT identity dependent manner. Genes that were upregulated in a 'HIF1A-specifc' manner ('HIF1A Up') and up- or downregulated in a 'HIF2A-specific' manner ('HIF2A Up' and 'HIF2A Down') were defined for each PT identity. Cells of each PT identity were then separated into subsets and scored for the corresponding set of 'HIF1A Up', 'HIF2A Up' and 'HIF2A Down' genes. Genotype- and time-dependent differences in HIFα isoform specific gene expression scores were then visualized for cells of all PT identities plotted together.

To evaluate the expression of 'HIF1A Early Up', 'HIF2A Early Up', 'HIF2A Adaptive Up' and 'HIF2A Adaptive Down' gene sets in clinical ccRCC samples, RNA sequencing data from tumor and matched normal samples was first obtained from TCGA. Expression data of each gene was scaled across tumor and normal samples of all cancer types to have a mean value of 0 and standard deviation of 1. Scaled expression values of genes were summed to derive a score for the aggregate expression of a particular gene set in each sample. Differences in median scores were tested with the Kruskal-Wallis test using the 'kruskal_test' function in the rstatix[74] package (v0.2.7) in R, with Bonferroni correction for multiple testing.

## Statistics and reproducibility

All experiments were performed with at least three biological replicates (mice) per experimental condition. Mice of both sexes were

included in every experiment. Where possible, mice from the same litter were used for both 'early' and 'late' experimental cohorts. Experiments were repeated in multiple mouse cohorts generated over several years to enhance reproducibility of the findings. No statistical method was used to predetermine sample size. No data were excluded from the analyses. The experiments were not randomized. The investigators were not blinded to allocation during experiments and outcome assessment.

IHC quantification data was analyzed using non-parametric methods. Differences in medians were tested by Kruskal-Wallis test with Dunn's multiple testing correction using Prism 10 (GraphPad Software, Boston, Massachusetts USA, www.graphpad.com). Differences in frequency distributions were tested using multivariate analysis of variance with the 'manova' function in R and the Wilk's statistic. Resulting $p$ values were adjusted for multiple testing using Bonferroni's correction.

For DE testing, $p$ values were calculated using the Wald test in the 'deseq' function of the DESeq2 (v1.44.0)[70] package in R, against the null hypothesis that the absolute difference in L2FC between two conditions was equal to 0, setting the parameters 'independentFiltering = FALSE' and 'alpha = 0.01'.

Spearman coefficients for correlation (ρ) between differential gene expression in different genotypes were calculated for each PT identity separately using the 'cor.test' function in R.

### Data visualization
Single-cell sequencing data was visualized with the in-built functions of pheatmap (v1.0.12)[75], ggvenn (v0.1.10)[76], and ggplot2 (v3.5.1)[77] packages in R (v4.3.2 and v4.4.1)[78]. Statistical analyses and relevant plots were also generated in Prism 10 (GraphPad Software, Boston, Massachusetts, USA, www.graphpad.com).

### Reporting summary
Further information on research design is available in the Nature Portfolio Reporting Summary linked to this article.

## Data availability
The scRNA-seq data for *Vhl*-null cells (ConKO and VKO) harvested early and late after recombination data used in this study are available in the Gene Expression Omnibus (GEO) database under accession code GSE253168. The scRNA-seq data generated in this study for VHKO, VEKO, and VHEKO mice harvested late after recombination are available in the GEO database under accession code GSE282887. Source data for IHC quantification analyses are provided with this paper.

## Code availability
R script used to analyze scRNA-seq data and generate plots is provided on GitHub and can be accessed at https://github.com/samvid-k/Lima_et_al_2025 [https://doi.org/10.5281/zenodo.16816717][79].

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

## Acknowledgements

Funding for the work was received from the Oxford Branch of the Ludwig Institute for Cancer Research (P.J.R). This work was also supported by the Francis Crick Institute, which receives its core funding from Cancer Research UK (FC001501), the UK Medical Research Council (FC001501), and the Wellcome Trust (FC001501) (P.J.R). S.K. was sponsored by a Christopher Welch Scholarship and the Clarendon Fund. A.B.B. was sponsored by a National Council for Scientific and Technological Development (CNPq) award scholarship (2023). We thank Ruddy Montandon, Ghada Ben Youssef, and Mohammed Islam for Flow Cytometry services (Wellcome Trust Center of Human Genetics, University of Oxford). We also thank all staff in the Functional Genetics Unit, University of Oxford for their support with animal husbandry.

## Author contributions

J.D.C.C.L, A.B.B, N.M, J.A, and S.K. performed mouse husbandry, tamoxifen administration, and harvests. J.D.C.C.L, A.B.B., and S.K. processed tissues and performed tdTomato IHC. J.D.C.C.L. performed RNA in situ hybridization. M.H. performed tdTomato-Ki67 dual IHC. M.H. and S.K. performed quantitative analysis of IHC data. J.D.C.C.L. and S.K. prepared scRNA-seq samples and libraries. R.L, D.R.M., and S.K. processed scRNA-seq data. S.K. analyzed scRNA-seq and RNA-seq data. D.R.M. and S.K. designed the scRNA-seq analysis. D.R.M. accessed and analyzed data from TCGA. C.W.P. provided expert insights and guidance to the project. P.J.R. conceptualized the study and obtained funding. D.R.M., J.A., P.J.R., and S.K. supervised the project. J.D.C.C.L., P.J.R., and S.K. wrote the manuscript. M.H., C.W.P., D.R.M., and J.A. reviewed and edited the manuscript.

## Competing interests

P.J.R. is a non-executive director of Immunocore Holdings PLC. The remaining authors declare no competing interests.
