## [Transparent Peer Review file · Nature Communications]

HIF α isoform specific activities drive cell-type specificity of *VHL*-associated oncogenesis

Corresponding Author: Dr Samvid Kurlekar

Version 0:

Reviewer comments:

Reviewer #1

(Remarks to the Author)

Lima et al report analyses of tagged, Vhl-deficient, renal epithelial cells. They use an elegant, previously reported, system where they now disrupt Hif1a and Hif2a to assess the functional role of these proteins downstream of Vhl. Whereas Vhl loss in the renal papilla, results in Hif1a dependent loss of cells, Vhl loss in the cortex, results in expansion that is co-dependent on Hif1a and Hif2a. Through amalgamated, sorted, single cell transcriptomic analyses of PT cells 4-12 months after Vhl inactivation, they show that (i) some genes appear to be independent of both Hif1a and Hif2a; (ii) 40%-50% of genes depend on either; (iii) Hif2a exerts greater control over gene expression than Hif1a; and (iv) Hif2a, but not Hif1a, is implicated in a delayed (so called adaptive program).

The manuscript is interesting even if somewhat broad and the authors deserve commendation for their efforts to make their report clear.

Main

The authors report that: VHEKO cells (in which both Hifa isoforms were inactivated) did not cluster with ConKO cells, suggesting that Vhl-null PT cells undergo transcriptomic changes over time even in the absence of Hif1/2a. This should be expanded upon. What is the function of these genes? Can anything be garnered from their function about putative effector pathways?

Gene expression analyses in 4b and 4c appear to focus on p values and should include magnitude of change.

The analyses about neighbors draws on inferences that may or may not be accurate and is less robust than the rest of the manuscript.

It would be interesting to see how this relates to human ccRCC and to a lesser extent, but more practical, murine ccRCC.

While the association of Hif1a with glycolysis is expected, Hif2a association with cellular structure and motility, secretion/transport, and extracellular matrix organization is less so. How do these processes contribute to ccRCC development? Are these genes putative direct targets?

While beyond the scope of this work, the authors would be poised to simultaneously ablate Vhl with Pbrm1 (or Bap1) and assess the relative contribution of Hif1a and Hif2a ablation to tumor development. Of course, this requires breeding additional alleles and there is already a fair amount of complexity. Perhaps one could explore ex vivo approaches initially.

As previously, the authors report A and B subtypes. What is their significance with respect to human disease?

Minor:

In Fig. 3b it may be helpful to include a bar for the genes that were independent.

There may be more intuitive ways to represent the data in 4d.

The statement in line 164 is not clear: 95.7% of cells were assigned as PT S1, S2, or S3 proximal tubular cell types, consistent with the PT restriction of Pax8-CreERT2 at this tamoxifen dose. Is it the dose, the timing or both?

Do the authors believe that there is significance to the relative location of VHEKO single cell clusters in the different PT regions respect to the other clusters on the UMAP?

UMAP S3A plot appears less dense?

Reviewer #2

(Remarks to the Author)

This study addresses the important question of trying to understand the earliest molecular events that contribute to the evolution of clear cell renal cell carcinoma. Herein the authors extend their recently published studies in which they developed an elegant genetic system that directly marks the inactivation of the Vhl tumour suppressor gene through simultaneous activation of a fluorescent reporter, permitting unambiguous identification of loss of Vhl function upon Cre activation. Here the HIF-1alpha or HIF-2alpha transcription factors were co-deleted (alone and together) with Vhl deletion in mouse renal epithelia, allowing assessment of the phenotypic and transcriptional contributions of the different HIF proteins to loss of Vhl-mediated events. Phenotypic analyses focused on differential negative selection of Vhl mutant cells within different regions of the kidney, proliferation and clonal expansion using quantifications of immunohistochemical stainings, while transcriptomic analyses took advantage of scRNA-seq and the author's previously published expertise in analysing the effects of Vhl deletion in different subtypes of proximal tubular cells. These analyses were all carried out at the highest scientific standards. The major conclusions of the study are that HIF-1a and HIF-2a are both required for initial proliferation and clonal expansion, that HIF-1a and HIF-2a have unique transcriptional targets as well as overlapping targets at early stages after gene deletion and that HIF-2a contributes predominantly to gene expression patterns that emerge at later timepoints after Vhl deletion. Overall the studies are very rigorously conducted and make an important contribution to the understanding of ccRCC evolution. At the same time, several open questions should be addressed in order to strengthen the conclusions that can be drawn from the data, as well as to interpret the relevance for different aspects of ccRCC biology.

1. It would be important to validate some of the early or late HIF1a and HIF2a dependent transcriptional targets that were identified by scRNA-seq in kidney sections by antibody staining or by RNA-scope.
2. The scRNA-seq analyses represent a major resource for the research community but their importance is somewhat "undersold" by the current level of analyses that are presented in the paper in Figures 4 and 5. It would be good to also present gene expression changes at the level of individual genes, rather than just as GSEA groups and average expression of collected marker genes, through single cell heatmaps or bubble plots to highlight the most interesting and cancer-relevant findings as well as give readers an impression of the magnitude of the changes for at least a selection of interesting genes in the various categories.
3. It would also be important to compare these gene expression results to the lists of differentially expressed genes that were identified in previously published experiments in which combinations of Vhl and Hif1a and Epas1 were deleted in cultured renal epithelial cells (Schönenberger et al. Cancer Research 2016) or in bulk RNA-seq of mouse Vhl/Trp53/Rb1 mutant tumours (Harlander et al Nat Med 2017). Both concluded that there are HIF-1a and HIF-2a uniquely regulated as well as overlapping targets. These comparisons would be important to further illustrate the power of the experiments that have been performed in the current study (combining the power of the genetic approach with scRNA-seq of cells isolated directly from the in vivo context vs the more artificial setting of cultured cells vs bulk tumour RNA-seq). It would be useful for the field to know which in vivo bone fide transcriptional responses might be also maintained in the artificial setting of cell culture (or not) as well as those early and late HIF-1a or HIF-2a transcriptional responses that arise following Vhl deletion and that are also represented (or not) in a mouse ccRCC model.
4. Based on these analyses, it would also be appropriate if the authors could elaborate in the discussion section about the potential different contributions of HIF-1a/HIF-2a regulated genes to different aspects of ccRCC biology based on the classes of genes that each regulate. For example, in the results section HIF-2a is stated as regulating genes "pertaining to cellular structure and motility, secretion and transport, and extracellular matrix organization" and "enriched significantly for assorted transmembrane transport and metabolic processes (Fig. 4c and Supplementary Fig. 8), suggesting an action on renal tubular differentiation.". These descriptions are currently a little vague. Some speculation about the potential contributions of these biological activities to ccRCC development would be of interest to readers.
5. The observation that HIF-2a induces a loss of differentiation at later timepoints is potentially very interesting and of relevance to ccRCC biology. More detail should be provided in this context. Which genes and gene classes are altered? This observation also raises the question of how a transcriptional activator (HIF-2a) causes loss of gene expression. Activation of a transcriptional repressor? Do the genes display enrichment of binding sites for proximal tubule-specifying transcription factors (eg HNF4?) or for HIF-2a? Given that loss of RNA expression is less reliably detected than increased expression in scRNA-seq, due to possible technical limitations of read depth and missingness, it would be very important to validate this loss of differentiation through orthologous approaches, ideally involving antibody staining or RNA-scope.
6. The conclusions about the roles of HIF-1a and HIF-2a in inducing clonal expansion and early proliferation should be compared to the conclusions that were made in several previous studies (in addition to the conclusions from the

Vhl/Trp53/Rb1 tumour model that are currently referenced in the paper). Schönenberger et al Cancer Research 2016 similarly showed that both HIF-1a or HIF-2a deletion decreased proliferation in Vhl and in Vhl/Trp53 deficient kidneys. The current study reaches a similar conclusion but importantly has the advantage of definitively identifying the Vhl null cells, emphasising the cell autonomous nature of the phenotype, which could not be definitively concluded in the previous study. Other studies also described proliferative phenotypes induced by the expression of constitutively stabilized mutants of HIF-1a (Fu et al Cancer Research 2011) or HIF-2a (Shietke et al PLOS One 2012). The work of Rankin et al Cancer Research 2006 also implicated both HIF-1a and HIF-2a in renal cyst formation following Vhl deletion in mouse kidneys, albeit through a phenotype that arose very late in life and at low penetrance.

Reviewer #3

(Remarks to the Author)

Nicely executed paper assessing the relative contribution of Hif1a and Epas1 to phenotype seen in Vhl deficient cells in a Pax8-Cre driven murine transgenic model with tdTomato permitting selection for Vhl null cells. The authors demonstrate different growth rate in various PT cellular subtypes as a function of HIF isoform loss and demonstrate that both isoforms are necessary for enhanced growth in cortical/outer medullary cells. They next identify isoform specific gene expression levels using scRNA showing co-clustering between VKO and VHKO but not with VEKO cells, and that Epas1 regulated a larger number of exclusive genes. Gene ontology analysis demonstrated distinct differences between Hif1a and Epas1, with Epas1 driven genes associated with PT differentiation. Finally the authors demonstrate that adaptive gene changes are more strongly dependent on Epas1 than on Hif1a.

This is a well written paper that provides new information on the relative roles of the two major HIF isoforms on gene expression and cellular development. While it does not, by the nature of our animal models, permit direct observation of oncogenic development, it provides some clues as to how oncogenesis could be encouraged. There are no major flaws in the overall layout of the paper.

Version 1:

Reviewer comments:

Reviewer #1

(Remarks to the Author)

I am satisfied with the comments and support going forward with publication.

Reviewer #2

(Remarks to the Author)

Thank you to the authors for the extensive series of new experiments and analyses that they have added to the revised version of the manuscript, which fully address all of the issues that I had raised in my review. I believe that these new results have improved the overall study and I congratulate the authors on this lovely piece of work.

Reply to Reviewers

We thank the reviewers for their comments and have answered them point by point. Our replies are **typed in blue**. Relevant changes have been made to the revised manuscript as indicated and are called out in **bold typeface**. Changes made to the main text, references, and figure legends have been highlighted in yellow in the submitted 'tracked changes' document.

Reviewer #1

Lima et al report analyses of tagged, *Vhl*-deficient, renal epithelial cells. They use an elegant, previously reported, system where they now disrupt *Hif1a* and *Hif2a* to assess the functional role of these proteins downstream of *Vhl*. Whereas *Vhl* loss in the renal papilla, results in *Hif1a* dependent loss of cells, *Vhl* loss in the cortex, results in expansion that is co-dependent on *Hif1a* and *Hif2a*. Through amalgamated, sorted, single cell transcriptomic analyses of PT cells 4-12 months after *Vhl* inactivation, they show that (i) some genes appear to be independent of both *Hif1a* and *Hif2a*; (ii) 40%-50% of genes depend on either; (iii) *Hif2a* exerts greater control over gene expression than *Hif1a*; and (iv) *Hif2a*, but not *Hif1a*, is implicated in a delayed (so called adaptive program).

The manuscript is interesting even if somewhat broad and the authors deserve commendation for their efforts to make their report clear.

Main:

The authors report that: VHEKO cells (in which both *Hifa* isoforms were inactivated) did not cluster with ConKO cells, suggesting that *Vhl*-null PT cells undergo transcriptomic changes over time even in the absence of *Hif1/2a*. This should be expanded upon. What is the function of these genes? Can anything be garnered from their function about putative effector pathways?

This is a discerning point, but we have reservations about drawing secure conclusions for three reasons.

First, in comparing VHEKO to ConKO cells, we cannot determine whether gene expression differences arise from *HIF* α -independent effects of *Vhl* inactivation, or some effects of *HIF* α isoform deletion that are independent of *Vhl* status, i.e., resulting from basal levels of active *HIF* α in ConKO cells.

Second, although our data indicated that in the relevant *Vhl/HIF* α genotype, most cells manifesting *Vhl* inactivation are also inactivated for the intended *HIF* α isoform, this overlap is unlikely to be totally complete (**Supplementary Fig. 1c**). Furthermore, since levels of *HIF* α transcripts are too low to be reliably detected in every cell in which the gene is in fact intact, we cannot use such data to 'curate' cell populations such as VHEKO in order to achieve complete accuracy of combined recombination. Therefore, we cannot discern truly *HIF* α -independent effects of *Vhl* inactivation from the effects of incomplete *HIF* α recombination in *Vhl*-null cells.

Third, at the level of individual genes, this problem is compounded by the difficulty in statistical proof of no effect i.e. the difficulty in proving this against the inevitable background of some noise. For these reasons, we applied a high stringency threshold for proof of an effect of HIF α inactivation so that identification of genes regulated by each HIF α isoform should be accurate. However, we are concerned about the level of accuracy we could achieve in defining a truly HIF α -independent effect.

We have, however, performed some additional analyses to further explore this point. First, we plotted differential expression in VKO *versus* ConKO against differential expression in VHEKO *versus* VKO cells (**revised Supplementary Fig. 5a**). This demonstrated a very strong anti-correlation across the population of genes, consistent with inactivation of *Vhl* leading to the expected activation of HIF. Importantly, genes that had not passed our threshold for HIF α -dependence (colored grey in **Supplementary Fig. 5a**) still exhibited this anti-correlation, suggesting that the expression of these genes is not truly independent of HIF α . Second, we identified genes whose *Vhl*-dependent regulation was the least affected by concomitant *Hif1a/Epas1* deletion in any PT identity and performed GO term over-representation analysis. This did not yield any significantly enriched pathways, which (if present) might have supported the categorization of these genes as being biologically correct. Taken together, we believe that the distinction between HIF α -dependent and seemingly non-dependent genes in our study may be a quantitative rather than qualitative difference.

In the revised manuscript we have revised our interpretation of this data and discussed this more thoroughly in response to the reviewer's point (**Lines 204-207, 210-224**), but we would prefer not to expand on the individual genes or categorize them as truly HIF α -independent for the reasons outlined.

The full dataset on differential expression and our assignment of HIF α isoform-specific dependence to each gene is supplied as a resource (**Supplementary Table 2**), enabling investigators to pick up on this point should they wish (for instance in the context of other data).

Gene expression analyses in 4b and 4c appear to focus on p values and should include magnitude of change.

As suggested, we have performed further analyses to determine the magnitude of alterations in gene expression. Specifically, for each enriched GO term, we identified member genes that were significantly regulated following *Vhl* inactivation in any PT identity. We then calculated the log₂-fold change (L2FC) for each those genes in VHEKO *versus* VKO and VEKO *versus* VKO cells across all PT identities in which the genes were regulated by *Vhl* inactivation. We then averaged the L2FC values for member genes within each GO term to provide a measure of the magnitude of change. In the revised manuscript, this is represented as a heatmap in **revised Fig. 4b and 4c**.

The analyses about neighbors draws on inferences that may or may not be accurate and is less robust than the rest of the manuscript.

We have performed additional analyses to enhance the robustness of this data. First, we have performed a comparative analysis of the ‘control’ genotypes, which indicate that they do not manifest an increase in the number of tdTomato-positive ‘neighbors’ of tdTomato-positive cells with time (**revised Supplementary Fig. 3d; Lines 146-148**). Second, we have performed multivariate statistical testing on the time-dependent change in the number of tdTomato-positive ‘neighbors’ of tdTomato-positive cells (**revised Fig. 2a; Lines 143-146**).

We should also point out that these analyses are intended to be complementary to the differential analyses of cell numbers, in that they provide additional (albeit indirect) evidence of an expansion of cells that is not dependent on having induced similar levels of initial recombination in animals subsequently analyzed at the early and late time points. We have also clarified this rationale in the revised manuscript (**Lines 136-140**).

It would be interesting to see how this relates to human ccRCC and to a lesser extent, but more practical, murine ccRCC.

Thank you. In order to relate our findings to human ccRCC and to other murine models of ccRCC we have performed:

1. Gene-set enrichment analysis on three published RNA-seq datasets¹⁻³ performed on mouse ccRCC *versus* normal cortex for the sets of HIF1A- or HIF2A-dependent ‘early’ and ‘adaptive’ *Vhl*-dependent genes defined in our manuscript (**revised Supplementary Fig. 11**).
2. Evaluation of the collective expression of HIF1A- or HIF2A-dependent ‘early’ and ‘adaptive’ *Vhl*-dependent genes as defined in our manuscript in human ccRCC *versus* normal renal tissue as assessed from The Cancer Genome Atlas (**revised Fig. 6c**).

These analyses have demonstrated that both HIF1A- and HIF2A-dependent genes constituting the ‘early’ and ‘adaptive’ transcriptional program in *Vhl*-null cells are consistently upregulated in murine and human ccRCC when compared to normal renal tissue. These new analyses have been included and briefly discussed in the revised manuscript (**Lines 332-344, 427-430**).

While the association of *Hif1a* with glycolysis is expected, *Hif2a* association with cellular structure and motility, secretion/transport, and extracellular matrix organization is less so. How do these processes contribute to ccRCC development? Are these genes putative direct targets?

Though we do not know exactly how all the over-represented pathways contribute to the development of ccRCC, several that are regulated in a HIF2A-dependent manner contribute to processes that have been identified as hallmarks of cancer. For instance, reorganization of the extra-cellular matrix (e.g., collagen genes, collagen prolyl hydroxylase genes) and altered cellular transport (e.g., fatty acid transporters, amino acid transporters) are known to occur in ccRCCs and may contribute to deregulated cellular energetics and loss of homeostasis in growth⁴.

The reviewer also asks whether these are putative direct targets (we assume of HIF). Accurate distinction of direct and indirect targets of HIF, particularly HIF2A, is not straightforward (as is the case with many other transcriptional cascades). HIF2A is largely an enhancer-binding transcription factor i.e. it binds at a distance from its target genes⁵. In this circumstance direct action cannot be simply inferred from proximity. Accurate definition of direct targets in chromatinized DNA therefore requires careful structural and functional analysis of each gene locus.

We agree that the contribution of each pathway to the development and progression of ccRCC and the extent to which each gene involved is directly regulated by HIF2A are interesting questions raised by our work, but we consider that complete analysis lies outside of the scope of the current manuscript. Nevertheless, in accordance with reviewer's suggestion we have revised the discussion to consider these issues (**Lines 403-408**).

While beyond the scope of this work, the authors would be poised to simultaneously ablate Vhl with Pbrm1 (or Bap1) and assess the relative contribution of Hif1a and Hif2a ablation to tumor development. Of course, this requires breeding additional alleles and there is already a fair amount of complexity. Perhaps one could explore *ex vivo* approaches initially.

Thank you. We agree that this should be interesting in future studies but is outside the scope of the current work. Generating the required 'multi-allele' genotypes is very complex and time-consuming and presentation of the analytical work would greatly complicate the current manuscript. Likewise, an *ex vivo* approach would be at odds with the main thrust of this work which is to assess the effects of gene interventions in an *in vivo* context where (as a result of the marking strategy) early and later changes can be compared. For instance, even if techniques were available that would allow primary renal cultures to be sustained in culture over several months, they would not necessarily capture the effects of these additional gene inactivation events on tumor formation and would potentially confuse presentation of the current manuscript. However, we do agree with the reviewer that our work opens this interesting question for future study.

As previously, the authors report A and B subtypes. What is their significance with respect to human disease?

In previous work we have shown that cells bearing an A subtype gene expression pattern are more likely to be proliferative in this model and, following *Vhl* inactivation, to exhibit greater upregulation of genes that are prognostic markers in human ccRCC such as *Vegfa* and *P4ha1*, than cells of the B subtype⁶. Cells of the A subtype also share gene expression patterns with *VCAM1*-positive cells in 'normal' human kidney samples that are postulated to be potential cells-of-origin for ccRCC⁷⁻¹⁰. This has suggested that cells of the A subtype may be more closely associated with ccRCC

development. At present we are unable to say more. In this manuscript, we have again identified these two populations and shown that they are present independently of HIF α -status.

Minor:

In Fig. 3b it may be helpful to include a bar for the genes that were independent.

For reasons mentioned in reply to the reviewer's first comment, we cannot accurately assign genes that do not pass our threshold for HIF α -dependence as 'HIF α -independent. We would therefore prefer not to alter this figure. However, we have provided differential gene expression data on genes that have not been called HIF α -dependent in **revised Supplementary Table 2**).

There may be more intuitive ways to represent the data in 4d.

We have tested multiple ways of representing this data – heatmaps, combining data across multiple PT identities, displaying individual genes etc. – and we believe our current representation to be the clearest representation of the result. Fig. 4d (now **revised Fig. 5a**) illustrates that in each PT cell type, *Vhl* inactivation results in decreased expression of known cell-type specific marker genes and that this diminution is abrogated largely by *Epas1* but not *Hif1a* inactivation. In response to Reviewer 2, we have also included analysis of *Epas1*-dependent loss of differentiation markers in proximal tubular cells by *in situ* hybridization (**revised Fig. 5b**). Together, we hope these revisions improve the representation of the data.

The statement in line 164 is not clear: 95.7% of cells were assigned as PT S1, S2, or S3 proximal tubular cell types, consistent with the PT restriction of Pax8-CreERT2 at this tamoxifen dose. Is it the dose, the timing or both?

This result refers to the specific tamoxifen regimen used (i.e. both dosing and timing). We have revised our manuscript and amended this line to say “under these experimental conditions” instead of “at this tamoxifen dose” to account for this effect (**Line 172**).

Do the authors believe that there is significance to the relative location of VHEKO single cell clusters in the different PT regions respect to the other clusters on the UMAP?

We are wary of interpreting the relative location of VHEKO cells compared to cells of other genotypes in the UMAP space. Generally, distance-based analysis of UMAP plots should be interpreted with caution, since they do not represent a linear analysis of the two-dimensional representation of high dimension data^{11,12}.

UMAP S3A plot appears less dense?

Yes, this is correct, S3 A cells are fewer in number (1,265 cells in total) than the other cell types. Since each dot represents a single cell, they will therefore appear less dense in the UMAP plot.

Reviewer #2

This study addresses the important question of trying to understand the earliest molecular events that contribute to the evolution of clear cell renal cell carcinoma. Herein the authors extend their recently published studies in which they developed an elegant genetic system that directly marks the inactivation of the Vhl tumour suppressor gene through simultaneous activation of a fluorescent reporter, permitting unambiguous identification of loss of Vhl function upon Cre activation. Here the HIF-1alpha or HIF-2alpha transcription factors were co-deleted (alone and together) with Vhl deletion in mouse renal epithelia, allowing assessment of the phenotypic and transcriptional contributions of the different HIF proteins to loss of Vhl-mediated events. Phenotypic analyses focused on differential negative selection of Vhl mutant cells within different regions of the kidney, proliferation and clonal expansion using quantifications of immunohistochemical stainings, while transcriptomic analyses took advantage of scRNA-seq and the author's previously published expertise in analysing the effects of Vhl deletion in different subtypes of proximal tubular cells. These analyses were all carried out at the highest scientific standards. The major conclusions of the study are that HIF-1a and HIF-2a are both required for initial proliferation and clonal expansion, that HIF-1a and HIF-2a have unique transcriptional targets as well as overlapping targets at early stages after gene deletion and that HIF-2a contributes predominantly to gene expression patterns that emerge at later timepoints after Vhl deletion. Overall the studies are very rigorously conducted and make an important contribution to the understanding of ccRCC evolution. At the same time, several open questions should be addressed in order to strengthen the conclusions that can be drawn from the data, as well as to interpret the relevance for different aspects of ccRCC biology.

It would be important to validate some of the early or late HIF1a and HIF2a dependent transcriptional targets that were identified by scRNA-seq in kidney sections by antibody staining or by RNA-scope.

Thank you. We have revised our manuscript to include *in situ* RNA hybridization-based validation of some of our scRNA-seq results (**revised Supplementary Fig. 6**).

The scRNA-seq analyses represent a major resource for the research community but their importance is somewhat "undersold" by the current level of analyses that are presented in the paper in Figures 4 and 5. It would be good to also present gene expression changes at the level of individual genes, rather than just as GSEA groups and average expression of collected marker genes, through single cell heatmaps or bubble plots to highlight the most interesting and cancer-relevant findings as well as give readers an impression of the magnitude of the changes for at least a selection of interesting genes in the various categories.

As suggested, we have revised our manuscript to highlight the magnitude of regulation of individual genes that undergo HIF α isoform-specific regulation (**revised Fig. 4c, 4d, and 4f**). Data on

the differential expression and HIF α -dependence for each gene has also been provided in **Supplementary Table 2**.

It would also be important to compare these gene expression results to the lists of differentially expressed genes that were identified in previously published experiments in which combinations of *Vhl* and *Hif1a* and *Epas1* were deleted in cultured renal epithelial cells (Schönenberger et al. Cancer Research 2016) or in bulk RNA-seq of mouse *Vhl/Trp53/Rb1* mutant tumours (Harlander et al Nat Med 2017). Both concluded that there are HIF-1a and HIF-2a uniquely regulated as well as overlapping targets. These comparisons would be important to further illustrate the power of the experiments that have been performed in the current study (combining the power of the genetic approach with scRNA-seq of cells isolated directly from the *in vivo* context vs the more artificial setting of cultured cells vs bulk tumour RNA-seq). It would be useful for the field to know which *in vivo* bone fide transcriptional responses might be also maintained in the artificial setting of cell culture (or not) as well as those early and late HIF-1a or HIF-2a transcriptional responses that arise following *Vhl* deletion and that are also represented (or not) in a mouse ccRCC model.

We thank the reviewer for this suggestion. Accordingly, we have revised our manuscript to relate the HIF α -dependent program entrained in *Vhl*-null cells to *Vhl*- and HIF α -dependent gene expression changes reported in primary cultures of proximal tubular cells¹³, murine models of ccRCC¹⁻³, and in patient ccRCC samples as provided in TCGA.

To provide a comparison of our findings with work described in these publications, we have used gene set enrichment analysis (GSEA) for the mouse models. These analyses demonstrate highly significant associations between genes identified at early and late timepoints in our model and the published work (**revised Supplementary Fig. 11; Lines 332-344**).

For the cell culture work, we did not have access to quantitative data in the published work, so we used GSEA to compare quantitation of the effects of HIF1A and HIF2A in our work with the classification of genes being either HIF2A- or HIF1A-regulated in the published work (**revised Supplementary Fig. 10; Lines 279-286**). This revealed a striking contrast between a clear association of HIF1A-dependence in both systems and essentially no such association for the genes defined as HIF2A-regulated in our study. Importantly, this suggests that the effects of HIF2A are substantially context dependent, supporting the use of *in vivo* models to test its effects.

Consistent with this, we took our defined lists of HIF1A- and HIF2A-specific genes that are part of the early and adaptive program in *Vhl*-null cells and tested the aggregate expression of these genes in ccRCC *versus* normal kidney tissues derived from TCGA. We found gene groups to be concordantly regulated in our data and that of TCGA (**revised Fig. 6c; Lines 332-344**).

We have discussed our findings in the revised manuscript (**Lines 427-430**).

Based on these analyses, it would also be appropriate if the authors could elaborate in the discussion section about the potential different contributions of HIF-1a/HIF-2a regulated genes to different aspects of ccRCC biology based on the classes of genes that each regulate. For example, in the results section HIF-2a is stated as regulating genes “pertaining to cellular structure and motility, secretion and transport, and extracellular matrix organization” and “enriched significantly for assorted transmembrane transport and metabolic processes (Fig. 4c and Supplementary Fig. 8), suggesting an action on renal tubular differentiation.”. These descriptions are currently a little vague. Some speculation about the potential contributions of these biological activities to ccRCC development would be of interest to readers.

Thank you. We have revised our manuscript to expand on the potential roles of HIF1A- and HIF2A-dependent transcriptional regulation to ccRCC (**Lines 257-259, 403-408**).

Many of the genes downregulated in a HIF2A-dependent manner code for transmembrane transporters, vesicular trafficking and metabolic enzymes that characterize the physiological function of proximal tubular cells – re-absorption of water and solutes from the glomerular filtrate. This is indicative of a loss of normal epithelial function of *Vhl*-null PT cells, which is commonly observed in oncogenesis. Therefore, we speculate that HIF2A might be driving an aggressive undifferentiated phenotype in ccRCC cells. Upregulation of collagen and collagen processing genes, metalloproteinases, and dysregulation of cytoskeletal and growth factor receptor genes in the same cells further suggest epithelial-to-mesenchymal cell-type transitioning, another hallmark of oncogenesis. These processes are coupled to upregulated glycolytic metabolism through the actions of HIF1A, which is a well-known metabolic adaptation in ccRCC and other cancers. We have outlined these potential connections to aspects of ccRCC biology in the revised discussion and also highlighted associations with gene expression in human ccRCC (see response to Reviewer 1, point 4).

The observation that HIF-2a induces a loss of differentiation at later timepoints is potentially very interesting and of relevance to ccRCC biology. More detail should be provided in this context. Which genes and gene classes are altered? This observation also raises the question of how a transcriptional activator (HIF-2a) causes loss of gene expression. Activation of a transcriptional repressor? Do the genes display enrichment of binding sites for proximal tubule-specifying transcription factors (eg HNF4?) or for HIF-2a? Given that loss of RNA expression is less reliably detected than increased expression in scRNA-seq, due to possible technical limitations of read depth and missingness, it would be very important to validate this loss of differentiation through orthologous approaches, ideally involving antibody staining or RNA-scope.

We have revised the manuscript to expand on the action of HIF2A in promoting dedifferentiation.

As suggested, we have validated the HIF2A-dependent loss of representative PT S1, PT S2, and PT S3 differentiation markers *in situ* by RNA hybridization (**revised Fig. 5b**).

We have also highlighted some of the individual genes that may contribute to this process (revised Fig. 4f).

As also suggested, we have utilized two bioinformatic tools – LISA and CHEA3 - to assess which transcription factors are predicted to regulate the sets of genes whose expression is induced in *Vhl*-null cells in a HIF1A- and HIF2A-dependent manner. This analysis revealed that HIF2A-dependent genes, but not HIF1A-dependent genes, are enriched for predicted binding sites for several transcription factors with known roles in proximal tubular or renal development, such as *HNF4A*, *HNF1B*, *TBX3* and *FOXA2*, consistent with an action of HIF2A on proximal tubular differentiation. The reviewer also asks about the mechanism by which HIF2A leads to repression of gene expression. Previous work indicates that HIF does not bind at the loci of down-regulated genes, suggesting that the mechanism is indirect (e.g., through activation of a specific repressor, altered chromatin accessibility). We could not identify a specific repressor in this work, but low-level expression could escape detection in the transcriptomic protocols we deployed. In the revised manuscript we have included data from the new analyses (revised Fig. 5c) and revised the discussion to cover these points (Lines 268-278).

The conclusions about the roles of HIF-1a and HIF-2a in inducing clonal expansion and early proliferation should be compared to the conclusions that were made in several previous studies (in addition to the conclusions from the *Vhl*/Trp53/Rb1 tumour model that are currently referenced in the paper). Schönenberger et al Cancer Research 2016 similarly showed that both HIF-1a or HIF-2a deletion decreased proliferation in *Vhl* and in *Vhl*/Trp53 deficient kidneys. The current study reaches a similar conclusion but importantly has the advantage of definitively identifying the *Vhl* null cells, emphasising the cell autonomous nature of the phenotype, which could not be definitively concluded in the previous study. Other studies also described proliferative phenotypes induced by the expression of constitutively stabilized mutants of HIF-1a (Fu et al Cancer Research 2011) or HIF-2a (Shietke et al PLOS One 2012). The work of Rankin et al Cancer Research 2006 also implicated both HIF-1a and HIF-2a in renal cyst formation following *Vhl* deletion in mouse kidneys, albeit through a phenotype that arose very late in life and at low penetrance.

We have referenced these works and expanded our discussion on HIF α -dependent proliferation in our revised manuscript (Lines 389-397). We appreciate the reviewer's comments that our study has added single-cell resolution to the analysis of *Vhl*-null cell proliferation. By performing our analysis in advance of the development of any morphological abnormalities in the kidney, we have also been able to identify that *Vhl* loss (and HIF α stabilization) is a primary driver of cell proliferation specifically in the renal cortex and outer medulla.

Reviewer #3

Nicely executed paper assessing the relative contribution of Hif1a and Epas1 to phenotype seen in Vhl deficient cells in a Pax8-Cre driven murine transgenic model with tdTomato permitting selection for Vhl null cells. The authors demonstrate different growth rate in various PT cellular subtypes as a function of HIF isoform loss and demonstrate that both isoforms are necessary for enhanced growth in cortical/outer medullary cells. They next identify isoform specific gene expression levels using scRNA showing co-clustering between VKO and and VHKO but not with VEKO cells, and that Epas1 regulated a larger number of exclusive genes. Gene ontology analysis demonstrated distinct differences between Hif1a and Epas1, with Epas1 driven genes associated with PT differentiation. Finally the authors demonstrate that adaptive gene changes are more strongly dependent on Epas1 than on Hif1a.

This is a well written paper that that provides new information on the relative roles of the two major HIF isoforms on gene expression and cellular development. While it does not, by the nature of out animal models, permit direct observation of oncogenic development, it provides some clues as to how oncogenesis could be encouraged. There are no major flaws in the overall layout of the paper.

We thank the reviewer for the accurate summary of our work and their positive assessment of our manuscript.

References

- 1 Harlander, S. *et al.* Combined mutation in Vhl, Trp53 and Rb1 causes clear cell renal cell carcinoma in mice. *Nat Med* **23**, 869-877 (2017). <https://doi.org:10.1038/nm.4343>
- 2 Hoefflin, R. *et al.* HIF-1alpha and HIF-2alpha differently regulate tumour development and inflammation of clear cell renal cell carcinoma in mice. *Nat Commun* **11**, 4111 (2020). <https://doi.org:10.1038/s41467-020-17873-3>
- 3 Nargund, A. M. *et al.* The SWI/SNF Protein PBRM1 Restrains VHL-Loss-Driven Clear Cell Renal Cell Carcinoma. *Cell Rep* **18**, 2893-2906 (2017). <https://doi.org:10.1016/j.celrep.2017.02.074>
- 4 Hanahan, D. & Weinberg, R. A. Hallmarks of cancer: the next generation. *Cell* **144**, 646-674 (2011). <https://doi.org:10.1016/j.cell.2011.02.013>
- 5 Smythies, J. A. *et al.* Inherent DNA-binding specificities of the HIF-1alpha and HIF-2alpha transcription factors in chromatin. *EMBO Rep* **20** (2019). <https://doi.org:10.15252/embr.201846401>
- 6 Kurlekar, S. *et al.* Oncogenic Cell Tagging and Single-Cell Transcriptomics Reveal Cell Type-Specific and Time-Resolved Responses to Vhl Inactivation in the Kidney. *Cancer Res* **84**, 1799-1816 (2024). <https://doi.org:10.1158/0008-5472.CAN-23-3248>
- 7 Lombardi, O. *et al.* Conserved patterns of transcriptional dysregulation, heterogeneity, and cell states in clear cell kidney cancer. *Cell Rep* **44**, 115169 (2025). <https://doi.org:10.1016/j.celrep.2024.115169>
- 8 Young, M. D. *et al.* Single-cell transcriptomes from human kidneys reveal the cellular identity of renal tumors. *Science* **361**, 594-599 (2018). <https://doi.org:10.1126/science.aat1699>
- 9 Muto, Y. *et al.* Single cell transcriptional and chromatin accessibility profiling redefine cellular heterogeneity in the adult human kidney. *Nat Commun* **12**, 2190 (2021). <https://doi.org:10.1038/s41467-021-22368-w>
- 10 Zhang, Y. *et al.* Single-cell analyses of renal cell cancers reveal insights into tumor microenvironment, cell of origin, and therapy response. *Proc Natl Acad Sci U S A* **118** (2021). <https://doi.org:10.1073/pnas.2103240118>
- 11 Chari, T. & Pachter, L. The specious art of single-cell genomics. *PLoS Comput Biol* **19**, e1011288 (2023). <https://doi.org:10.1371/journal.pcbi.1011288>
- 12 Sun, S., Zhu, J., Ma, Y. & Zhou, X. Accuracy, robustness and scalability of dimensionality reduction methods for single-cell RNA-seq analysis. *Genome Biol* **20**, 269 (2019). <https://doi.org:10.1186/s13059-019-1898-6>
- 13 Schonenberger, D. *et al.* Formation of Renal Cysts and Tumors in Vhl/Trp53-Deficient Mice Requires HIF1alpha and HIF2alpha. *Cancer Res* **76**, 2025-2036 (2016). <https://doi.org:10.1158/0008-5472.CAN-15-1859>